# Dynamical evolution of $CO_2$ and $H_2O$ on garnet electrolyte elucidated by ambient pressure X-ray spectroscopies

Nian Zhang [1,2,7], Guoxi Ren[2,7], Lili Li[3,7], Zhi Wang[4], Pengfei Yu[2], Xiaobao Li[2], Jing Zhou [3], Hui Zhang [1,2] ✉, Linjuan Zhang [3], Zhi Liu [5] ✉ & Xiaosong Liu [2,5,6] ✉

Garnet-type $Li_{6.5}La_3Zr_{1.5}Ta_{0.5}O_{12}$ (LLZO) is considered a promising solid electrolyte, but the surface degradation in air hinders its application for all-solid-state battery. Recent studies have mainly focused on the final products of the LLZO surface reactions due to lacking of powerful in situ characterization methods. Here, we use ambient pressure X-ray spectroscopies to in situ investigate the dynamical evolution of LLZO surface in different gas environments. The newly developed ambient pressure mapping of resonant Auger spectroscopy clearly distinguishes the lithium containing species, including LiOH, $Li_2O$, $Li_2CO_3$ and lattice oxygen. The reaction of $CO_2$ with LLZO to form $Li_2CO_3$ is found to be a thermodynamically favored self-limiting reaction. On the contrary, the reaction of $H_2O$ with LLZO lags behind that of $CO_2$, but intensifies at high pressure. More interestingly, the results provide direct spectroscopic evidence for the existence of $Li^+/H^+$ exchange and reveal the importance of the initial layer formed on clean electrolyte surface in determining their air stability. This work demonstrates that the newly developed in situ technologies pave a new way to investigate the oxygen evolution and surface degradation mechanism in energy materials.

The increasing desire for higher capacity and better safety energy storage technology is fostering the revolution of lithium-ion batteries (LIBs)[1–3]. Among all possible anode materials, Li metal is considered as the ultimate choice to boost the energy density in LIBs because of its high theoretical capacity (3869 mA h $g^{-1}$) and low redox potential[1,4]. However, the growth of Li dendrites and the severe volume changes limit the applications of Li metal batteries[5–7]. Non-flammable solid-state electrolytes (SSEs) with high shear modulus and $Li^+$ transference number have potential to solve these problems[8,9]. Among various types of SSEs, garnet-based solid electrolyte $Li_7La_3Zr_2O_{12}$ is highly

promising due to its high ionic conductivity (up to 1 mS $cm^{-1}$ at 25 °C), high electrochemical stability towards Li metal, the feasibility of mass production and possible storage in air[10,11]. Nevertheless, previous findings have pointed out that a lithiophobic $Li_2CO_3$ layer forms on the surface of LLZO as a result of exposure to $CO_2$ and $H_2O$ in air[12,13]. This thin $Li_2CO_3$ layer, moreover, have shown to surprisingly governs the interface property, leading to poor interfacial contact and high interfacial impedance between LLZO and electrode materials[14].

Different pathways for the surface reaction have been proposed to describe the actual reaction between LLZO and air. The density

[1]Shanghai Synchrotron Radiation Facility, Shanghai Advanced Research Institute, Chinese Academy of Sciences, Shanghai 201204, China. [2]State Key Laboratory of Functional Materials for Informatics, Shanghai Institute of Microsystem and Information Technology, Chinese Academy of Sciences, Shanghai 200050, China. [3]Shanghai Institute of Applied Physics, Chinese Academy of Sciences, Shanghai 201204, China. [4]Institute of Semiconductors, Chinese Academy of Sciences, Beijing 100089, China. [5]Center for Transformative Science, Shanghai Tech University, Shanghai 201210, China. [6]National Synchrotron Radiation Laboratory, University of Science and Technology of China, Hefei 230026, China. [7]These authors contributed equally: Nian Zhang, Guoxi Ren, Lili Li. ✉e-mail: huizhang@mail.sim.ac.cn; liuzhi@shanghaitech.edu.cn; xsliu19@ustc.edu.cn

functional theory (DFT) calculations indicate that, thermodynamically, $CO_2$ can react directly with LLZO to produce $Li_2CO_3$[13]. However, the reaction seems kinetically slow as only a negligible amount of $Li_2CO_3$ was observed experimentally on the surface of LLZO after long-term exposure to anhydrous air[15]. In another highly probable pathway, $H_2O$ first reacts with LLZO to form LiOH through $Li^+/H^+$ exchange, then a substantial part of LiOH transforms into $Li_2CO_3$ after exposure to $CO_2$[16]. Spontaneous $Li^+/H^+$ ion exchange is believed not to change the cubic garnet structure of LLZO, but blocks $Li^+$ transmission channel[16,17]. Therefore, a core−shell structure is proposed, comprising a garnet core surrounded by a proton-rich garnet shell and a LiOH/$Li_2CO_3$ outer layer. Up to now, this core-shell structure has only been studied by destructive characterization techniques such as argon ion sputter etching[18], and the reaction mechanism of $Li_2CO_3$ formation has not been fully understood. Thus, it is of great significance to develop in situ techniques with depth-profiling capability to comprehensively probe the initial reactions at gas/solid interface, which can help us understand the different thermodynamic and kinetic processes for $CO_2$ and $H_2O$ on LLZO surface, and eventually provide guidance for avoiding the formation of $Li_2CO_3$ and building an excellent interface.

Synchrotron-based core-level X-ray spectroscopies, including X-ray photoelectron spectroscopy (XPS) and X-ray absorption spectrum (XAS), have been widely used to monitor the gas/solid interface in catalysis and environmental science[19-21]. Recent developed in situ spectroscopy techniques can be utilized to investigate the reaction between gas and the solid surface with elemental and chemical sensitivities. Additionally, for soft-X-rays, XAS in total-electronic-yield (TEY) mode can support a detection depth of ~10 nm, while signals in auger-electron-yield (AEY) mode usually come from a depth of ~3 nm[22]. Thus, simultaneous detection of XAS in both TEY and AEY mode along with XPS allows us to obtain an in-depth analysis of the chemical evolution on the surface and sub-surface. However, due to the poor air stability and very similar spectroscopic fingerprints, lithium containing species such as $Li_2O$, $Li_2O_2$, LiOH, $Li_2CO_3$ and LLZO are very difficult to identify, thus very limited studies have combined all these surface methods in ambient pressure for energy materials characterization until now.

In this paper, the initial surface chemistry and evolution mechanism of LLZO in $H_2O$ and $CO_2$ were in situ investigated by ambient pressure X-ray photoelectron and absorption spectroscopy. In particular, the newly developed ambient pressure mapping of resonant Auger spectroscopy (AP-mRAS) method clearly identifies lattice oxygen (LLZO) and various surface oxygen species such as LiOH, $Li_2O$ and $Li_2CO_3$, and the in situ depth-profiling technology can deduce the flow direction of lithium. We find that $CO_2$ reacts directly with LLZO thermodynamically, but the reaction is limited by surface active sites and a lack of oxygen supply. The reaction of $H_2O$ with LLZO needs a relatively higher pressure than $CO_2$ through $Li^+/H^+$ exchange. However, the reaction is more intense and continuous, forming LiOH. Comparing LLZO with $Li_{1.5}Al_{0.5}Ge_{1.5}P_3O_{12}$ (LAGP), we found that the characteristics of the interface layer initially formed on the surface and the stability of the material structure may be the decisive factors for its resistance to air degradation.

## Results

### Clean LLZO surface obtained by low temperature treatment

The LLZO pellet suffers severe surface degradation after exposure to air. Figure 1a, b shows the surface and cross-section morphology of the LLZO pellet after exposure to air for 2 months. Large amounts of amorphous structures are visible around the voids of LLZO pellet, assignable to $Li_2CO_3$ which is evidenced by the presence of C and O in energy-dispersive X-ray spectroscopy (EDS) analysis in Fig. 1c[23,24]. After polishing by sandpaper and wiping by alcohol, the LLZO particles can be clearly observed on the surface with a small amount of residual debris Fig. 1d, e. The results also indicate that completely

clean LLZO surface cannot be obtained only by physical polishing. The X-ray diffraction (XRD) pattern of the aged LLZO pellet is displayed in Fig. 1f. The diffraction peaks match well with the cubic garnet phase with no other impurities, which means that the phase of the bulk LLZO is barely changed during the surface chemical evolution in air, and the $Li_2CO_3$ contamination may have an amorphous structure.

The schematic diagram of the entire in situ experimental processes in this work are shown in Fig. 2. A clean LLZO surface is necessary for monitoring of the degradation mechanism. To achieve a clean LLZO surface, the polished LLZO pellet was annealed at 350 °C in vacuum for 30 min and then annealed in $1 × 10^{-6}$ mbar $O_2$ at 350 °C for 30 min, as shown in Fig. 2a−e. During the vacuum annealing process, in the range of 30 to 350 °C under ultra-high vacuum (UHV) condition, XPS, mass spectrum and XAS in AEY and TEY mode were carried out to characterize the surface chemical evolution. The schematic and the photo of the instrument are shown in Fig. 3a, b. XPS and XAS in AEY mode were conducted by the Scienta Hipp3 analyzer, while the XAS in TEY mode was measured by a pico-ammeter[25,26].

The polished LLZO pellet was mounted onto a sample holder under ambient air (which took about 15 min) and were pumped into the instrument. Thus, a strong signal of $Li_2CO_3$ and weak signals of La and Zr could be observed in XPS spectra as shown in Fig. 3c and Supplementary Fig. 1. The peak at 284.6 eV in C 1$s$ spectrum can be assigned to the C-C ($sp^2$) bond of the surface contamination[27]. The O K edge AEY and TEY spectra in Supplementary Fig. 2 are consistent with our XPS results that the surface of LLZO is mainly $Li_2CO_3$ before annealing. A small signal of LLZO is present at ~531.9 eV in the AEY spectrum, which is slightly weaker than in the TEY spectrum with a larger investigation depth of ~10 nm, indicating that the thickness of the contamination layer is several nanometers.

In order to determine whether the surface $Li_2CO_3$ can be completely removed by vacuum annealing, the O 1$s$ and C 1$s$ XPS spectra were measured at 350 °C. Only the peak of LLZO at around 529 eV labeled as O(lattice) can be detected in the O 1$s$ XPS spectrum as shown in Supplementary Fig. 3a. The results indicate that a clean LLZO surface without other O containing species can be achieved by low-temperature vacuum annealing[28]. However, after cooling down the LLZO pellet to room temperature, the peak of $Li_2CO_3$ appears again, as evident from Fig. 3c, indicating the clean LLZO surface may be very sensitive to residual $CO_2$[29] and can form $Li_2CO_3$ on the surface even at a pressure below $1 × 10^{-8}$ mbar. $Li_2CO_3$ is difficult to be observed in the AEY and TEY spectra of annealed LLZO at RT in Supplementary Fig. 2, indicating that very little $Li_2CO_3$ is generated on the surface during the cooling process. Thus, a clean surface of LLZO with very small amount of $Li_2CO_3$ is obtained by vacuum annealing.

Further, we systematically investigated the influence of temperature rise on the ratios of O(lattice)/O($CO_3^{2-}$) and C($CO_3^{2-}$)/C(C-C), delineated in Fig. 3d. The decomposition temperature of the surface $Li_2CO_3$ can be identified to be around 300 °C, which is much lower than the reported reaction temperature (over 620 °C for the reaction $Li_2CO_3 → Li_2O + CO_2$)[30,31]. No signal of $Li_2O$ was detected in the O 1$s$ XPS and O K edge mRAS data of annealed LLZO as shown in Supplementary Fig. 4. To probe the decomposition reaction at low-temperature, the decomposition gaseous products were detected by in situ mass spectrum, and the results are exhibited in Fig. 3e. Both $CO_2$ and $H_2O$ were simultaneously observed as the decomposition products around 300 °C. Thus, the actual reaction at the LLZO surface during the vacuum annealing process could be: $xLi_2CO_3 + Li_{6.5-2x}H_{2x}La_3Zr_{1.5}Ta_{0.5}O_{12} → Li_{6.5}La_3Zr_{1.5}Ta_{0.5}O_{12} + xH_2O + xCO_2$ as depicted in Fig. 3f[18,32]. However, surface contaminated carbon species cannot be removed by vacuum annealing, leading to the relatively weak signal of carbonate in C 1$s$ as seen in Fig. 3c. In order to prevent the strong signal of contaminated C-C $sp^2$ from affecting the observation of the evolution of carbonate, we then annealed the sample in $1 × 10^{-6}$ mbar

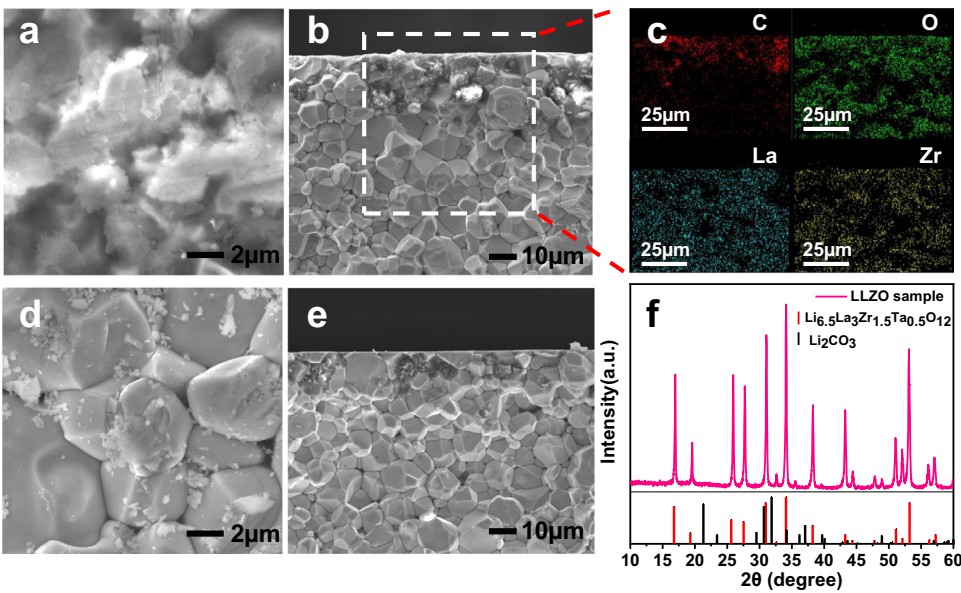

**Fig. 1 | Surface degradation of LLZO after exposure to air. a** Surface and **b** cross-section scanning electron microscope (SEM) images of aged LLZO pellet. **c** EDS elemental mappings of the aged sample. **d** Surface and **e** cross-section SEM images of polished LLZO pellet. **f** XRD patterns of the aged LLZO pellet.

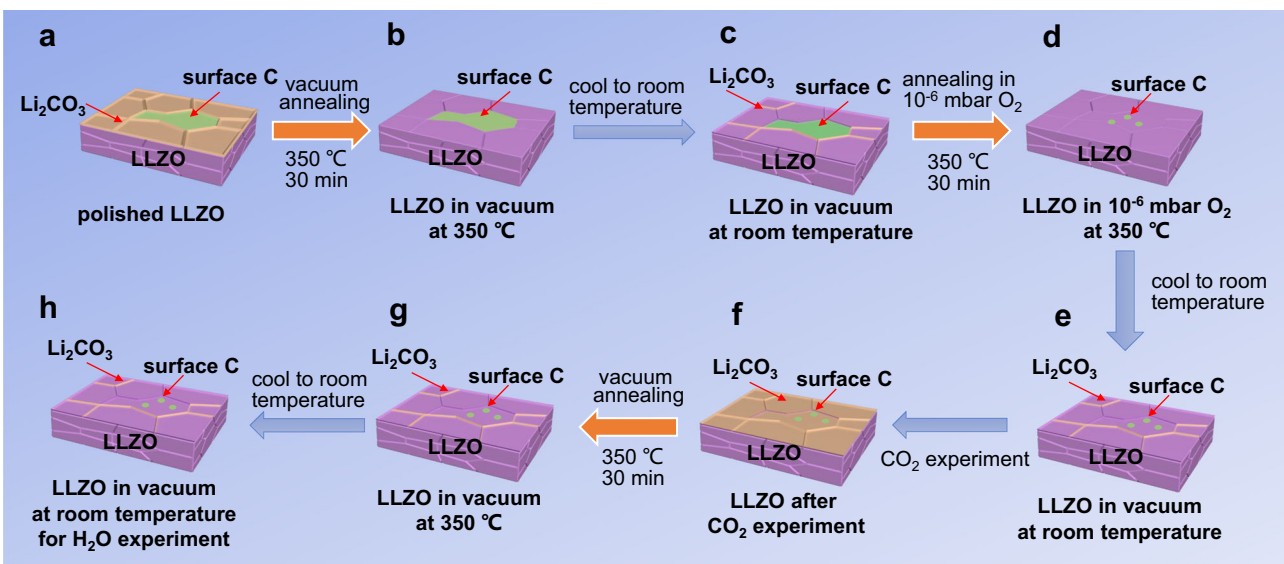

**Fig. 2 | Schematic diagram of the entire in situ experimental processes in this work. a, b** Polished LLZO pellet was vacuum annealing at 350 °C for 30 min to remove surface $Li_2CO_3$, but surface contaminated carbon species cannot be removed. **c** During the cooling process, a small amount of $Li_2CO_3$ will be generated on the surface of LLZO. **d,e** Most of the surface contaminated carbon species can be removed by annealing in $1 \times 10^{-6}$ mbar $O_2$ at 350 °C for 30 min. **f–h** The clean LLZO surface then be used to investigate the reaction with $CO_2$ and $H_2O$.

$O_2$ at 350 °C for 30 min. After annealing, the signals of other elements remained basically unchanged, and the signal of contaminated carbon species significantly decreased.

## Clear identification of lithium containing species using AP-mRAS and XAS

Due to the poor air stability and very similar spectroscopic fingerprints, lithium containing species are very difficult to identify by surface sensitive characterization methods[33,34]. Here, using the newly developed AP-mRAS method, we can carefully identify lattice and surface oxygen in lithium containing species before studying the reaction mechanism. The mechanism and characteristics of mRAS compared with mapping of resonant inelastic X-ray scattering (mRIXS) are shown in Supplementary Fig. 5, the incident photon energy is

scanned across the absorption edge, and the emitted Auger electrons at each resonant energy are further resolved in kinetic energy (KE).

Li metal was in situ scraped using a wobble-stick with sharp blade as shown in Supplementary Fig. 4. By utilizing near ambient pressure technology, we can obtain mRAS of pure $Li_2O$, LiOH and $Li_2CO_3$ for comparison. Figure 4a compares annealed LLZO with $Li_2O$, LiOH and $Li_2CO_3$, which shows completely different characteristics. The results support that the surface of annealed LLZO is clean. The AP-mRAS spectra of LLZO surface at different states are shown in Fig. 4b: LLZO surface after physical polishing mainly contains $Li_2CO_3$ on the surface, which displays a vertical symmetrical feature at photon energy (h$\nu$) of 533.7 eV. In the kinetic energy (KE) direction, the strongest point of $Li_2CO_3$ is located at around 512.2 eV and the intensity extends to both high and low KE directions. After surface treatment, clean LLZO

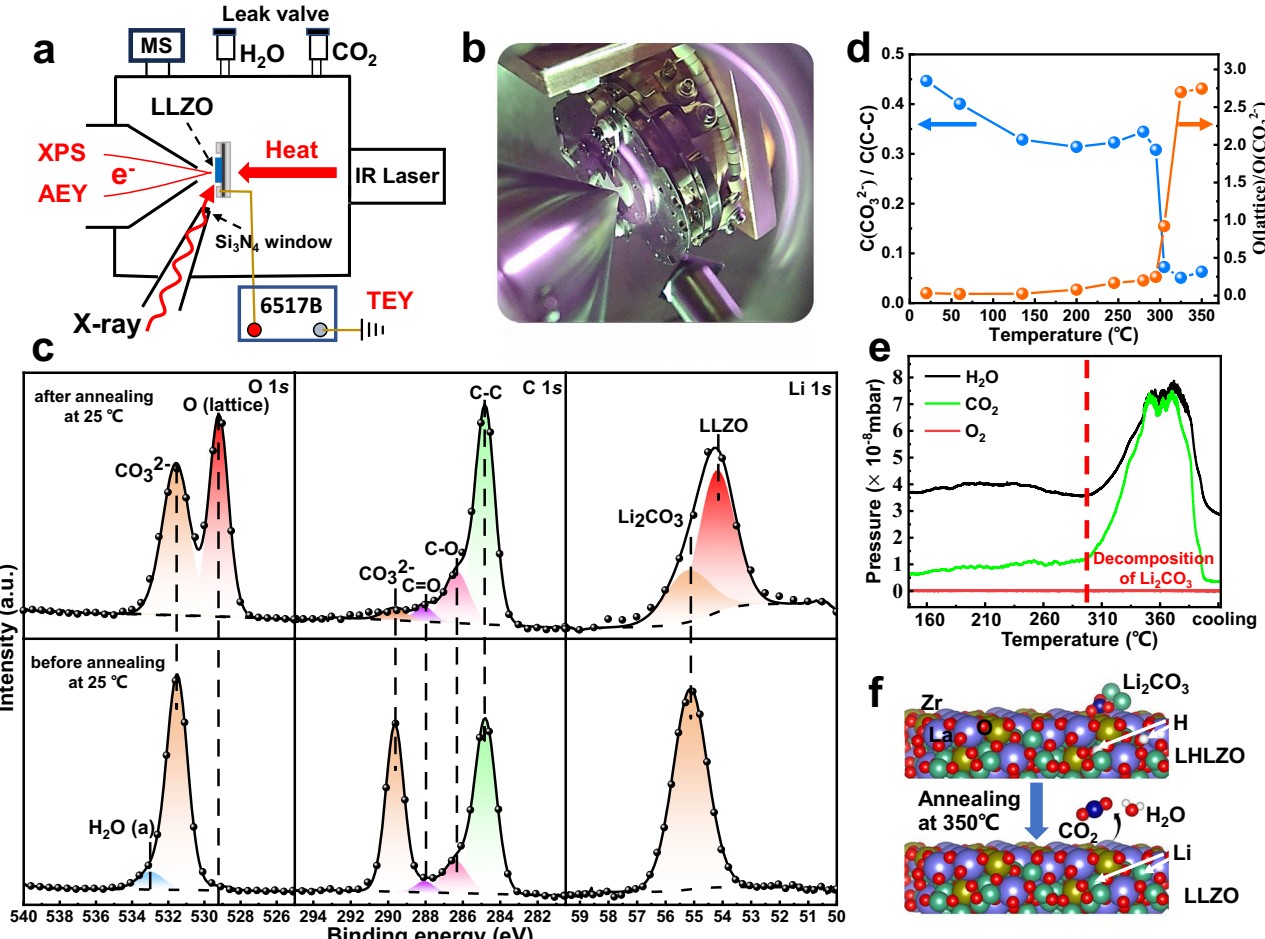

**Fig. 3 | Clean LLZO surface obtained by low temperature vacuum annealing.**
**a** Schematic and **b** real instrument for the operando ambient pressure experiment.
**c** XPS spectra of the LLZO pellet before and after vacuum annealing at room temperature. **d** The influence of temperature rise on the ratios of O(lattice)/$O(CO_3^{2-})$ and $C(CO_3^{2-})$/C(C-C). **e** In situ mass spectrum results in the temperature range of 160 – 370 °C. $CO_2$ and $H_2O$ are simultaneously observed to be the decomposition products at around 300 °C. **f** The schematic illustration of reaction of $x Li_2CO_3 + Li_{6.5-2x}H_{2x}La_3Zr_{1.5}Ta_{0.5}O_{12} \rightarrow Li_{6.5}La_3Zr_{1.5}Ta_{0.5}O_{12} + x H_2O + x CO_2$ at the LLZO surface during the vacuum annealing process.

surface shows an oblique spot at photon energy around 531.9 eV. The inclination of the spot depends on the *d* orbital property that hybridized with the O 2*p* orbital. Furthermore, clean LLZO surface shows a localized feature in KE direction with a much smaller photon energy which is quite different from $Li_2O$, which confirms the surface contamination is fully wiped out. Two parallel structures with the same binding energy are Ta 4*f* peaks, which come from the doped Ta in LLZO. For the Ap-mRAS of LLZO after the introduction of 1 mbar $CO_2$, the coexistence of LLZO and $Li_2CO_3$ features are observed, which proves that only a small amount of $Li_2CO_3$ (<3 nm) is produced on the surface. Compared with the signal of $Li_2CO_3$, the signal of LiOH is much stronger after 0.5 mbar $H_2O$ is introduced. LiOH also displays a vertical symmetrical feature at photon energy of 532.8 eV and its intensity extends along the KE direction. $Li_2CO_3$ and LiOH show very close KE values, which is 1.5 eV lower than that of LLZO as shown in Fig. 4d. These results reveal that the inclination, localization and the KE position of the intensity center in the two-dimensional spectrum can be used to accurately identify lattice oxygen (LLZO) and surface oxygen in lithium containing species such as LiOH, $Li_2O$ and $Li_2CO_3$. Combined with in situ mass spectrometry, AP-mRAS is a potential method to study anionic redox behavior in various cathode and catalytic materials. After the identification of lithium containing species, we can use the in situ ambient pressure technologies to study the specific reaction kinetics and thermodynamic process, which will be detailed in the following.

## The detailed reaction process of clean LLZO surface with $CO_2$

The advanced synchrotron radiation ambient pressure technology helped us to slow down the rapid reactions of $CO_2$/$H_2O$ on clean LLZO surface, so that we can clearly observe the thermodynamic and kinetic reaction processes. The kinetics of the reaction of clean LLZO with $CO_2$ was investigated by in situ ambient pressure APXPS and XAS measurements to achieve a depth-profiling analysis of the surface reaction. Figure 5a, b shows the O 1*s* and C 1*s* APXPS spectra of the LLZO at increasing pressure of $CO_2$. During the reaction, the reaction product is almost pure $Li_2CO_3$ which is illustrated in Supplementary Fig. 6. The peak intensity of $CO_3^{2-}$ at around 531.5 eV enhances with the increase of $CO_2$ pressures from $1 \times 10^{-6} - 1 \times 10^{-2}$ mbar, implying that $CO_2$ can rapidly react with LLZO to form $Li_2CO_3$ even at low $CO_2$ pressure. However, the peak intensity of $CO_3^{2-}$ remains almost unchanged upon increasing the pressure of $CO_2$ from $1 \times 10^{-2}$ to 1 mbar. The results reveal that the reaction of $CO_2$ on the LLZO surface may be restricted by surface active sites or oxygen supply from the sub-layer.

The O K edge AEY and AP-mRAS spectra of the LLZO pellet at increasing $CO_2$ pressures are displayed in Fig. 5c and Supplementary Fig. 7. The variation in the intensity of the $Li_2CO_3$ peak is consistent with that of XPS; the peak increases from $1 \times 10^{-6}$ to $1 \times 10^{-2}$ mbar and then stabilizes from $1 \times 10^{-2}$ to 1 mbar. We normalized the AEY data at different $CO_2$ pressures to achieve a clearer internal reaction image and then subtracted the UHV data ($I_{pressures}$-$I_{UHV}$). The difference results are presented in Fig. 5d, where the spectrum at $1 \times 10^{-6}$ mbar

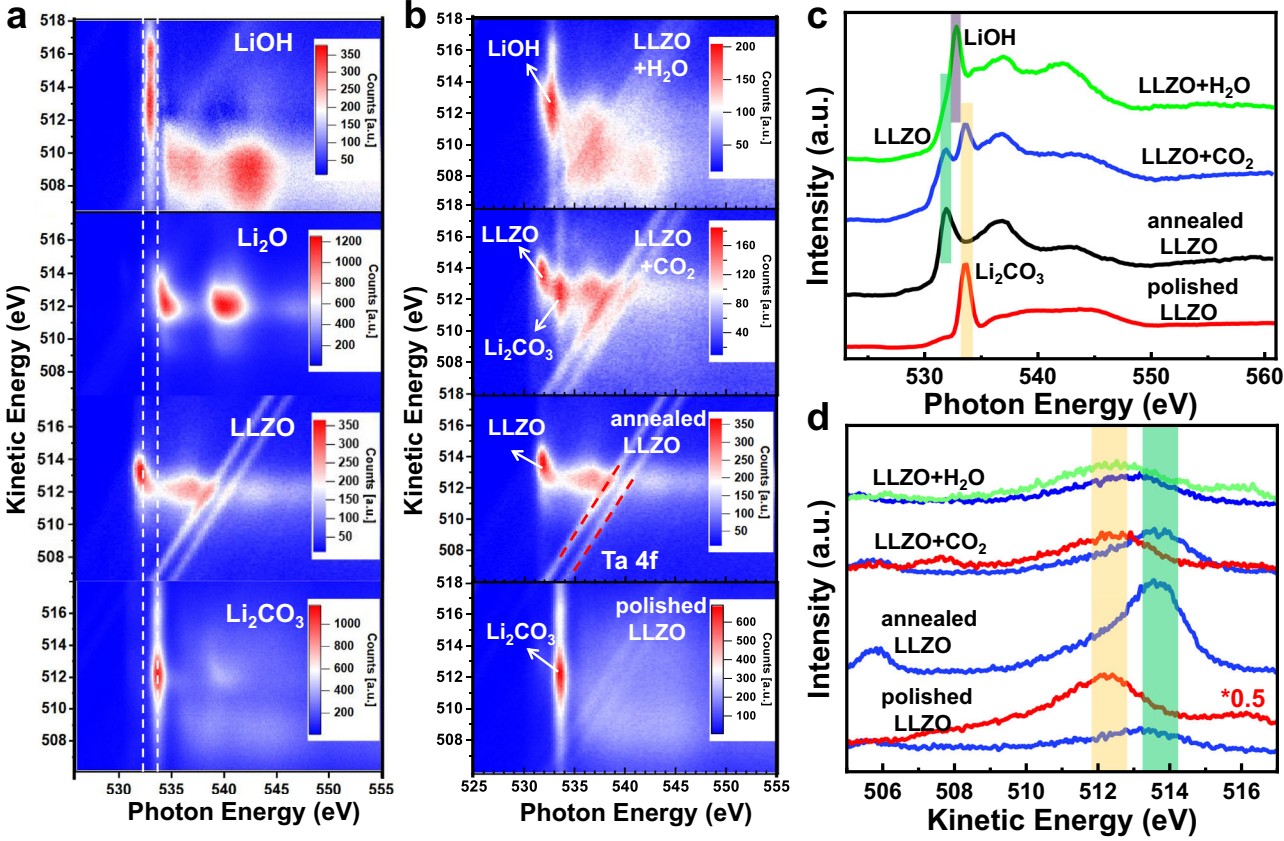

**Fig. 4 | The mRAS and XAS spectra of pure LiOH, Li$_2$O, Li$_2$CO$_3$ and LLZO surface at different states. a** The color-coded mRAS spectra comparison of clean LLZO with pure LiOH, Li$_2$O, Li$_2$CO$_3$. **b** The color-coded mRAS spectra comparison of LLZO surface at different states, corresponding to the process **a**, **e**, **f**, **h** in flowchart Fig. 2. The pressure of CO$_2$ and H$_2$O are 1 mbar and 0.5 mbar separately. **c** Distribution of

LLZO surface at different states in photon energy direction. **d** Distribution of LLZO surface at different states in kinetic energy direction. LLZO (blue line) is extracted from h$\nu$ = 531.9 eV, Li$_2$CO$_3$ (red line) is extracted from h$\nu$ = 533.7 eV and LiOH (green line) is extracted from h$\nu$ = 532.8 eV.

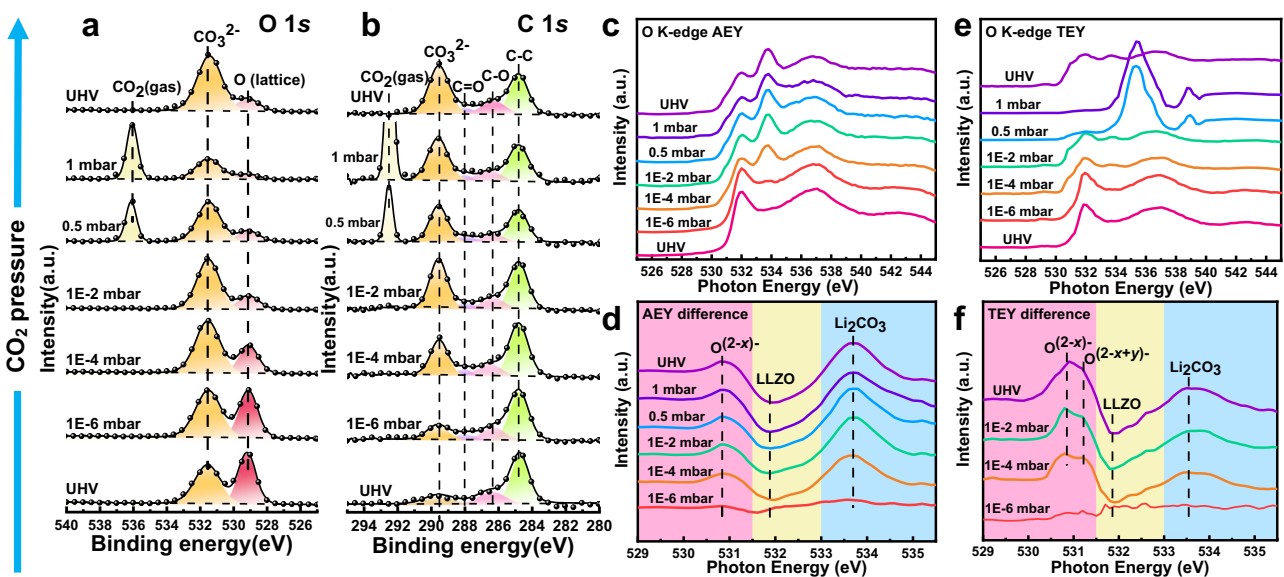

**Fig. 5 | The evolution of clean LLZO surface with the introduction of CO$_2$ studied by APXPS and APXAS. a** The variation of O 1$s$ and **b** C 1$s$ APXPS spectra at increasing CO$_2$ pressure from UHV to 1 mbar. **c,d** AEY spectra and difference images at different CO$_2$ pressures. A new feature at 530.8 eV appears, which can be

assigned to the high valence oxygen labeled as O$^{(2-x)-}$. **e,f** TEY spectra and difference images at different CO$_2$ pressures. Except the O$^{(2-x)-}$, a new peak at 531.2 eV is observed above $1 \times 10^{-4}$ mbar, attributed to the O$^{(2-x+y)-}$.

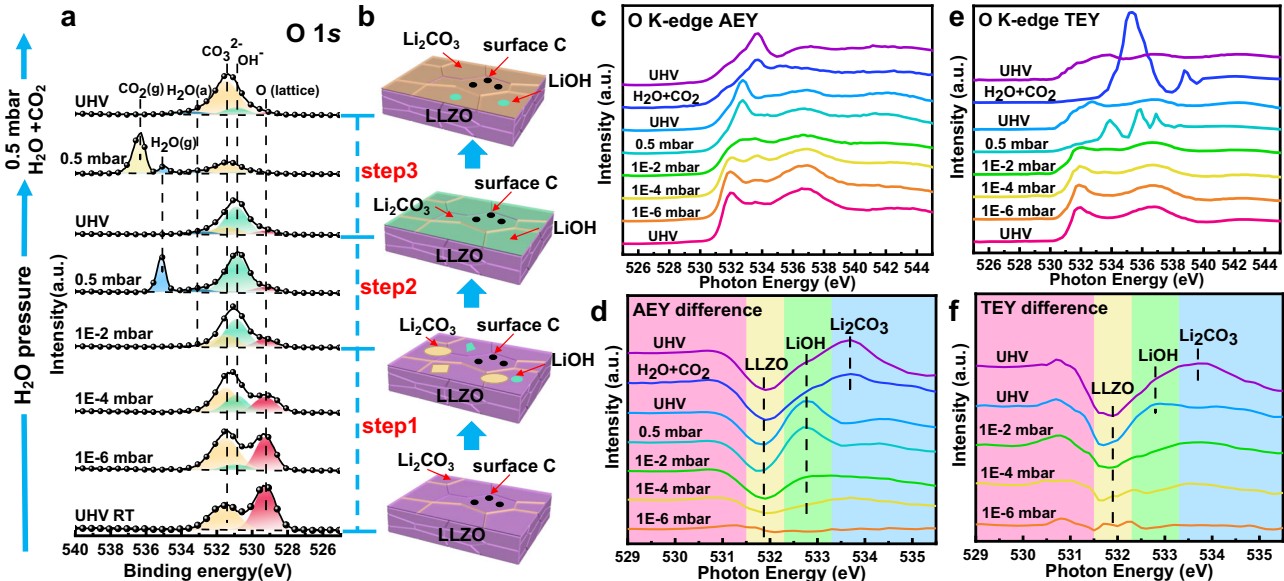

**Fig. 6 | The evolution of LLZO surface with the introduction of H$_2$O studied by APXPS and APXAS. a** The variation of O 1$s$ APXPS spectra at increasing H$_2$O pressure from UHV to 0.5 mbar, followed by the introduction of a mixture gas of 0.5 mbar H$_2$O and 0.5 mbar CO$_2$. The high binding energy peaks appear at high pressure are the gas peaks of H$_2$O and CO$_2$. The peak at around 533.3 eV is the adsorbed H$_2$O on LLZO surface. **b** Evolution diagram of LLZO surface reaction products. **c,d** AEY spectra and difference images at different H$_2$O and CO$_2$ pressures. No O$^{(2-x)-}$ feature is found in the difference spectra at 530.8 eV. **e,f** TEY spectra and difference images at different H$_2$O and CO$_2$ pressures. The slight fluctuations below 532 eV in the difference spectra may come from the signal noise.

CO$_2$ is close to the horizontal line, meaning that the reaction is insignificant at this pressure. When the CO$_2$ pressure is escalated to $1 \times 10^{-4}$ mbar, the signal of Li$_2$CO$_3$ intensifies significantly while the peak intensity of LLZO dramatically decreases, indicating the formation of Li$_2$CO$_3$ on the LLZO surface. Noteworthy, a new feature at -530.8 eV appears, which can be assigned to high valence O$^{(2-x)-}$ caused by the Li extraction[35–37]. The O$^{(2-x)-}$ formation is induced when the lithium in the sub-layer are pulled to the surface to form Li$_2$CO$_3$ because there is not enough lithium around the oxygen atom on the surface. Our TEY results with a higher detective depth of -10 nm are shown in Fig. 5e. Due to the interference of signal from CO$_2$, effective TEY signals from the LLZO surface could not be obtained above 0.5 mbar. Besides the 530.8 eV signal, a new peak at 531.2 eV is observed in difference spectra in Fig. 5f above $1 \times 10^{-4}$ mbar, which can be assigned to the O$^{(2-x+y)-}$. The observation of high valence O$^{(2-x)-}$ and O$^{(2-x+y)-}$ at different depths suggests the existence of Li gradient in the sub-layer of LLZO after exposure to CO$_2$, confirming our hypothesis that Li from the sub-layer is pulled to the surface to form Li$_2$CO$_3$, while oxygen remain in their original positions.

The variations of Li 1$s$, Zr 3$d$, and La 4$d$ APXPS spectra at increasing CO$_2$ pressure are shown in Supplementary Fig. 8. The Li 1$s$ peak shifts to higher binding energy as the CO$_2$ pressure rises, indicating the formation of Li$_2$CO$_3$ on the surface. No noticeable spectral line shape changes are seen in Zr 3$d$ and La 4$d$ spectra. Thus, the possible reaction between CO$_2$ and clean LLZO surface can be described as following: the CO$_2$ molecules may easily adsorb on particular oxygen sites on the surface of LLZO at very low CO$_2$ pressure. Subsequently, the lithium in the sub-layer are pulled to the surface to form Li$_2$CO$_3$ through the reaction: Li$_{6.5}$La$_3$Zr$_{1.5}$Ta$_{0.5}$O$_{12}$ + $x$CO$_2$ → Li$_{6.5-2x}$La$_3$Zr$_{1.5}$Ta$_{0.5}$O$_{12-x}$ + $x$Li$_2$CO$_3$. In contrast, LLZO is not a good oxygen ionic conductor at room temperature, the oxygen in the sub-layer have difficulty migrating to the surface. When the above definite oxygen sites are fully occupied, CO$_2$ cannot react with LLZO further, resulting in the Li$_2$CO$_3$ layer being only 1–3 nm thick. Consequently, the CO$_2$ reaction pathway is considered kinetically slow which is coincided with the negligible amount of Li$_2$CO$_3$ formed on garnet pellets after exposure to dry air[18,38,39].

**The detailed reaction process of clean LLZO surface with H$_2$O**

After the CO$_2$ experiment, the LLZO pellet was treated using the same annealing method as shown in Fig. 2f–h. Interestingly, the Li$_2$CO$_3$ could not be removed completely by vacuum annealing as the signal of Li$_2$CO$_3$ was still observable in the O 1$s$ XPS spectrum at 350 °C in Supplementary Fig. 9a. The results confirm the following decomposition reaction of Li$_2$CO$_3$ at 350 °C: $x$Li$_2$CO$_3$ + Li$_{6.5-2x}$H$_{2x}$La$_3$Zr$_{1.5}$Ta$_{0.5}$O$_{12}$ → Li$_{6.5}$La$_3$Zr$_{1.5}$Ta$_{0.5}$O$_{12}$ + $x$H$_2$O + $x$CO$_2$. Thus, the decomposition reaction of Li$_2$CO$_3$ would require a higher temperature if the hydrogens in the sub-layer LLZO are fully removed.

Furthermore, the reaction of H$_2$O with the LLZO pellet was studied by APXPS and APXAS. The O 1$s$ XPS spectra and the product evolution diagram at increasing H$_2$O pressure are displayed in Fig. 6a, b. Surprisingly, the peak intensity of CO$_3^{2-}$ at 531.5 eV enhances with increasing H$_2$O pressure from UHV to $1 \times 10^{-4}$ mbar, probably due to the gas path can only be cleaned to $1 \times 10^{-7}$ mbar and there is a small amount of residual CO$_2$. The results also indicate that the reaction of Li$_{6.5}$La$_3$Zr$_{1.5}$Ta$_{0.5}$O$_{12}$ + $x$CO$_2$ → Li$_{6.5-2x}$La$_3$Zr$_{1.5}$Ta$_{0.5}$O$_{12-x}$ + $x$Li$_2$CO$_3$ at clean LLZO surface may be a thermodynamically favorable route compared to the reaction of LLZO with H$_2$O, which is consistent with a just published article by Grey's group[40]. As the pressure of H$_2$O increases to 0.5 mbar, the signal of OH$^-$ at 530.9 eV[41] almost covers the peak of CO$_3^{2-}$, revealing that H$_2$O mostly reacts with LLZO at 0.5 mbar and the reaction products are much more than that of CO$_2$. Moreover, we compared the variation of O(CO$_3^{2-}$)/O(lattice) and O(OH$^-$)/O(lattice) during the introduction of CO$_2$ and of H$_2$O, respectively, and the corresponding results can be viewed in Supplementary Fig. 10. The reaction with CO$_2$ shows a deceleration process, while that of H$_2$O accelerates as the pressure increases. These results corroborate that the reaction of CO$_2$ is a thermodynamically favorable route compared with that of H$_2$O at low pressure. Next, we introduced a mixture gas of 0.5 mbar H$_2$O and 0.5 mbar CO$_2$, most of the surface LiOH is converted into Li$_2$CO$_3$ as shown in Fig. 6a and Supplementary Table 1.

The O K edge AEY and AP-mRAS spectra, shown in Fig. 6c and Supplementary Fig. 11, give a detailed analysis of the surface reaction, and the obtained results are consistent with the APXPS. The intensity of Li$_2$CO$_3$ enhances as the pressure of H$_2$O increases from UHV to

$1 \times 10^{-4}$ mbar. However, the signal of LiOH at 532.8 eV nearly covers the signal of LLZO over 0.5 mbar, indicating that the reaction of $H_2O$ is more severe than $CO_2$. Since no effective TEY spectra could be collected at the 0.5 mbar $H_2O$ pressure in Fig. 6e, we collected the spectrum when returning from 0.5 mbar to UHV instead. Even though the peak of LiOH is significantly intense, the LLZO signal can still be detected in the TEY spectrum at UHV, indicating that the thickness of LiOH layer at 0.5 mbar $H_2O$ is slightly <10 nm, much thicker than the $Li_2CO_3$ layer formed at the pressure of 1 mbar $CO_2$. Notably, signals of $O^{(2-x)-}$ and $O^{(2-x+y)-}$ at around 530.8 eV were absent in all of the AEY and TEY difference spectra shown in Fig. 6d, f, signifying no valence change of oxygen in LLZO sub-layer after exposure to $H_2O$. These results establish the existence of $Li^+/H^+$ exchange, i.e., the $H^+$ fill the vacancies after the $Li^+$ are pulled to the surface. After introducing a mixture gas of 0.5 mbar $H_2O$ and 0.5 mbar $CO_2$, the peak of $Li_2CO_3$ in both AEY and TEY spectra is unambiguously observed, indicating the existence of the surface reaction: $2LiOH + CO_2 \rightarrow Li_2CO_3 + H_2O$.

## Comparison of LLZO with LAGP

In order to find the possible determining factor for the air stability of electrolyte materials, we used the same in situ methods to study LAGP, which is an air stable solid electrolyte. The LAGP ceramic pellet was removed from its sealed packaging and placed directly into the test chamber from air without polishing. The signals of P, Ge and O can still be seen on the surface, indicating that LAGP has much better air stability than LLZO. Vacuum annealing at 350 °C hardly changes the composition of the surface. Then, we annealed the sample in $1 \times 10^{-6}$ mbar $O_2$ at 350 °C for 30 min. As shown in the Supplementary Fig. 12, it can be seen that the surface C of the sample at 350 °C significantly decreases, while the signals of O and P are greatly enhanced. However, when the sample was cooled to room temperature, the signal of C increased again while the signal of O and P decreased. No such carbonization phenomenon is observed on the surface of LLZO.

We conducted APXPS experiments on clean LAGP and the results are shown in Supplementary Fig. 13. LAGP cannot react with $CO_2$ and $H_2O$ even at a high pressure of 0.5 mbar. By comparing two electrolyte materials, we infer that the stability of LAGP may come from the hydrophobic carbon layer generated on the surface and the stability of $PO_4$ structure[42]. Due to the stability of surface O, it is difficult for LAGP to directly react with $CO_2$ to generate $Li_2CO_3$. On the contrary, LLZO surface can react directly with $CO_2$ to form a $Li_2CO_3$, which is a hydrophilic layer, may cause the occurrence of subsequent severe reactions. The results indicate that the hydrophilicity and hydrophobicity of the initial layer formed on the clean electrolyte surface, as well as the stability of the surface O structure, may play an important role in determining the air stability of solid electrolyte materials.

## Discussion

The human factor in peak fitting has always been a problem in XPS data processing, especially in the overlapped O 1s XPS data. Thus, the mRAS and AEY method are developed to assist XPS in species identification. The mRAS and AEY method can be implemented at any synchrotron radiation XPS end-station without requiring additional hardware. The mRAS and AEY has a detection depth slightly higher than XPS which not only can assist XPS in species identification, but also provide deep analysis abilities. In addition, its identification of species is also clearer and more intuitive because additional dimensions can result in more fingerprint features just like mRIXS as shown in Supplementary Fig. 5. Due to the dimensional improvement, mRIXS in O K edge has played an important role in the study of oxygen redox in cathode materials. The detection efficiency of mRAS is much higher than that of mRIXS, and the detection is also not affected by the gas and electrochemical environment. Thus, the mRAS method combined with near ambient pressure depth-profiling characterization methods possess the substantial potential and, therefore, should be widely popularized to

study the air stability and the thermodynamics and kinetics process of gas/solid interface in energy materials.

In conclusion, in situ ambient pressure depth-profiling techniques were initiatively used to elucidate the dynamical evolution of LLZO pellet with $CO_2$ and $H_2O$. Low-temperature vacuum annealing helped us to obtain a clean LLZO surface through the reaction: $xLi_2CO_3 + Li_{6.5-2x}H_{2x}La_3Zr_{1.5}Ta_{0.5}O_{12} \rightarrow Li_{6.5}La_3Zr_{1.5}Ta_{0.5}O_{12} + xH_2O + xCO_2$, where the H travel from LHLZO on the surface of LLZO. The APXPS, AP-mRAS and APXAS results experimentally prove that the reaction of $CO_2$ with LLZO to form $Li_2CO_3$ is a thermodynamically favored path. However, the $CO_2$ reaction is restricted by the hindered oxygen supply from the sub-layer, affording only 1–3 nm thick $Li_2CO_3$ layer. The driving force of the reaction leads to the formation of a lithium gradient in the sub-layer of LLZO. Moreover, $Li^+/H^+$ exchange was directly observed as no lithium gradient appeared during the reaction. This exchange is more intense when $H_2O$ reaches a higher pressure. Our results give a precise mechanism of the initial reaction of LLZO with $CO_2$ and $H_2O$ and reveal the initial layer may play an important role in determining the air stability of solid electrolyte materials.

## Methods

### Materials and characterizations

The starting materials of $Li_2CO_3$ (Alfa Aesar, 99.9%), $La_2O_3$ (Alfa Aesar, 99.9%), $ZrO_2$ (Alfa Aesar, 99.5%) and $Ta_2O_5$ (Aladdin, 99.5%) were mixed in stoichiometric amounts with 15 mol% $Li_2CO_3$ in excess. The mixture was ball-milled in 2-propanol for 12 h with agate balls in an agate vial, and then dried and heated in air at 1150 °C for 12 h. Then the ball-milling was repeated once, and the powder was sieved with a mesh number of 600 to obtain fine particles. The pellets were made by hot-pressing of the as-prepared LLZO powder in a flowing argon atmosphere at a temperature of 1050 °C under a constant pressure of 50 MPa for 1 h. The size of pellet for the operando experiment was 0.8 mm in thickness and 12 mm in diameter[23]. 0.3 mm thick LAGP ceramic pellet was purchased from Hefei Kejing Material Technology Co., Ltd. The X-ray powder diffraction (XRD) was performed using a Bruker D8 advance with Cu Kα radiation ($\lambda = 1.54178$ Å). Sintered and densified LLZO pellet was carried out from the glove box, then dry polished in air progressively using polishing paper with grit number from 800 to 1500, and the surface was wiped by alcohol. After polishing treatment, the polished and aged pellets were quickly transferred to the test chamber for the surface and cross section scanning electron microscopy (SEM) images, using a JEOL JSM-7800F field emission microscope.

### Operando ambient pressure experiment

In situ annealing, ambient pressure mapping of resonant Auger electronic spectroscopy (AP-mRAS), ambient pressure X-ray photoelectron spectroscopy (APXPS) and X-ray absorption spectroscopy (APXAS) were carried out at BL02B at the Shanghai Synchrotron Radiation Facility (SSRF). The pellets were brought to the SSRF through an aluminum plastic bag sealed in a glove box. Before transferred to UHV chamber, the LLZO pellet was polished in air by sandpapers with grit number from 800 – 1500 to achieve parallel faces. The polishing thickness was sufficient to ensure that the surface contamination layer is completely polished off and then the surface was wiped by alcohol. The polished pellet was mounted onto a sample holder under ambient air (which took about 15 min) and were pumped into the instrument. LAGP and Li metal were pumped into the chamber without polishing process. Li metal was in situ scraped using a wobble-stick with sharp blade surface. For the vacuum annealing process, an infrared laser heater (912 nm wavelength, PREVAC) was used for heating from backside of the sample holder. The temperature was monitored by the K type thermocouple attached onto the LLZO pellet. The temperature was increased at the rate of 3 °C/min from room temperature to 370 °C and then kept at 350 °C for 30 min before

spectroscopic measurements. The mass spectrum was collected by using MKS e-Vision 2 residual gas analyzer (RGA) installed on the analysis chamber. After conducting high-temperature measurements, LLZO pellet was naturally cooled to room temperature at a base pressure $\sim 1 \times 10^{-8}$ mbar for subsequent in situ characterization. After vacuum annealing, the LLZO pellet was annealed in $1 \times 10^{-6}$ mbar $O_2$ at 350 °C to reduce surface contaminated carbon.

The beamline 02B was a bending magnet beamline providing photons with energy range from 40 – 2000 eV. The photon flux was about $10^{11}$ photons/s and the energy resolving power was up to 13000. The C 1 $s$, O 1 $s$, Li 1 $s$, Zr 3$d$ and La 3$d$ XPS spectra were all collected at the photon energy of 650 eV with a step size of 0.1 eV using the Hipp-3 electron energy analyzer (Scienta Omicron). The photon energy was calibrated by a gold foil on the sample holder and the binding energy was calibrated by the C 1 $s$ peak on LLZO at 284.6 eV. The APXAS data of O K-edge were collected simultaneously by using Auger electron yield (AEY) mode with investigation depth $\sim$3 nm and total electron yield (TEY) mode with the penetration depth $\sim$10 nm. The photon energy step was set at 0.2 eV for O K-edge mRAS and TEY experiments and the collection time of each mRAS mapping was about 12 min. The window of kinetic energy for O K-edge mRAS was set to $512 \pm 7$ eV. All the spectra have been normalized to the beam flux measured by the upstream gold mesh. The stainless-steel tubes for $CO_2$ (99.999%) and water vapor were baked in vacuum conditions and then flushed several times using high-purity $CO_2$ and water vapor, respectively, before introducing the gases into the chamber. High-purity $CO_2$ and water vapor were introduced into the reaction chamber by controlling two independent all-metal leak valves (VACGEN) as shown in Fig. 3a. The gas pressure was read by a capacitance film vacuum gauge (PFEIFFER CMR 363) attached on the chamber. All spectra were collected after the pressure had stabilized for 15 min. For the experiments of a mixture of $H_2O + CO_2$, the water vapor of 0.5 mbar was firstly maintained, then $CO_2$ was introduced until the total pressure stabilized at 1 mbar.

## Data availability

The data supporting the findings of this study are available within the article and its Supplementary Information. Additional data are available from the corresponding authors on request. Source data are provided with this paper.

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

## Acknowledgements

This work is supported by the National Key R&D Program of China (Grant No. 2019YFA0405601 X.S.L.), the National Natural Science Foundation of China (Grant Nos. 11905283 N.Z.) and the National Key R&D Program of China (Grant No. 2022YFA1503801 Z.L., No. 2022YFE0198500 H.Z.). The authors thank BL02B of Shanghai Synchtrotron Radiation Facility supported by National Science Foundation of China under contract No. 11227902 Z.L.. The authors thank prof Yong Han for disscusion.

## Author contributions

N.Z., G.X.R., X.B.L. and H.Z. conducted the experiments. L.L.L, J.Z. and L.J.Z. participated in discussions and established the possible mechanism model. N.Z., G.X.R., H.Z. and Z.L. are responsible for the in situ ambient pressure depth-profiling X-ray spectroscopies and analysis. N.Z., Z.W., P.F.Y., H.Z. and X.S.L. designed the project, analyzed the results, and wrote the manuscript.

## Competing interests

The authors declare no competing interests.
