## [Peer Review File · Nature Communications]

REVIEWER COMMENTS

Reviewer #1 (Remarks to the Author):

This manuscript described a study using multiple X-ray spectroscopy method (in-situ) with different probing depths to study the solid electrolyte material. Theoretical understanding based on DFT was discussed. The reviewer found the manuscript at its current stage is not sufficient to be considered for publication at Nature Comms., primarily due to the below concerns. Major revision is required before potential reconsideration/re-submission.

Concern 1. Detailed characterization via multiple surface sensitive (in-situ) probes of garnet based solid electrolyte had already been very well documented and reaction mechanism/path was already understood. Noted the author adopted resonant Auger spectroscopy, which was not commonly used for garnet solid electrolyte surface. The rationale for using this method is not clear and the reviewer did not find sufficient amount of insights that was not previously discussed/reported.

Concern 2. The reviewer found the discussion remained shallow with limited new understanding on the reaction mechanism and steps.

Despite the concerns, the reviewer did find the merits of the proposed experimental flow. The reviewer recommends that this method/experimental flow could be quite powerful only if applied to different solid electrolyte surface/interface systems for a comparative study. New insights could be delivered from differences associated with chemistry, composition, micro-structure and etc. It is a bit unfortunate that those comparisons are not included in this manuscript.

Some additional technical comments are provided below.

Fig 3d. The polished LLZO surface seemed fully covered with Li_2CO_3 with a very weak signal of Ta. Was the polishing properly done? Or the sample got exposed to air/moisture/ CO_2 during sample transfer? A properly polished LLZO should be free of Li_2CO_3 and at least show strong La, Ta, Zr signals.

It is a very misleading discussion on the point of oxygen supply at the sub-layer when the authors discuss CO_2 reaction with LLZTO garnet surface. First, the LLZTO + dry CO_2 reaction, if this could kinetically occur, is a solid-gas reaction. There are no available "ions" that can possibly participate. It is only Li and

O atoms sitting on the crystal-graphic site on the surface and near the surface that reacts with CO₂. Therefore the reaction formula $2\text{Li}^+ + \text{O}_2^- + \text{CO}_2 \rightarrow \text{Li}_2\text{CO}_3$ is very misleading and author should consider use proper defect chemistry note to rewrite the reaction. Third, LLZTO is not a good oxygen ionic conductor at least at the temperature of this study and mostly "oxygen ion" moves via oxygen vacancies. The fact that dry CO₂ reaction with LLZTO to form Li₂CO₃ has nothing to do with so called migration of oxygen ion from the sublayer to surface. This physical image is not realistic and would entail crystal structure and micro-structure change, based on which experimental results, the author rule out the possibility of CO₂ react with Li₂O at the surface of LLZTO?

That theoretical explanation of adsorption capacity is quite misleading too. First, what is the adsorption capacity, mole of CO₂/H₂O molecular per LLZTO surface area? Second, how is this adsorption capacity (semi)-/quantified such that the authors can possible compare that capacity is high or low? Third, how is the capacity (mostly likely a thermodynamic term) determines the reaction rate (a kinetic term?) Lastly, there exists numerous report that LLZTO reacts with water almost immediately while the reactions with CO₂ in dry conditions are very slow. How does that reconcile author's statement that CO₂ is more active than H₂O on LLZTO surfaces. Could the author extract based on the very rich experimental observation a reason rate of constant to support the discussion?

The reviewer found Li gradient as result of reaction with CO₂ is not well supported. The author claimed the system may have an oxygen stoic difference. This itself is not supporting Li gradient since LLZTO charge neutrality by change in oxygen stoic does not have to be compensated by Li stoichiometry, especially when the off stoic variation is very small value. Additional experimental proof is needed to confirm the gradient of Li.

Reviewer #2 (Remarks to the Author):

The authors present an interesting paper and try to shed some light on a relevant question in the filed of Garnet electrolyte research. They employ some uncommon experimental methods, which promises new insights not possible before.

However, some of the methods employed involve a quite complex physics and may be misleading. The authors should give more details on some of those experiments, especially where the conclusions drawn are unexpected or contradicting to other findings.

The abstract already has some strange flaws, raising some scepticism:

Line 22: „The air stability of solid state electrolyte should be overcome for successful commercialization of all-solid-state Li-ion battery.” This doesn't make sense.

Line 23: “However, the surface degradation mechanism of solid state electrolyte is still ambiguous due to the lack of powerful in situ gas/solid interface characterization tools.” A surface degradation mechanism cannot be ambiguous.

Line 48: “recent findings” with a citation from 2014. Well, that's not very recent.

There is a recent report from the Grey group, <https://pubs.acs.org/doi/full/10.1021/acsenergylett.3c01042>, that deals with the same questions and uses some overlapping methods. The authors need to consider this paper in their arguments. Specifically, in that paper the change in the O1s at around 300 °C is linked to protonation, while no change in Lithiumcarbonate can be found. Their interpretation is much better supported by experimental evidence imo.

The partial pressure dependent measurements are new and interesting! But the description is lacking some experimental details. How where the gas flows controlled, how where the partial pressures determined?

The authors also need to give more experimental details with regard to sample preparation and handling, since this is so crucial in the case of LLZO. How where the samples transferred between experiments? When where samples broken to create fresh surface? How where the surfaces prepared, polished, etc.?

The authors claim that the XPS-spectra in Fig 1 of the supplementary show signs of Li₂CO₃ after annealing. However, Lithiumcarbonate shows a very distinguished signal in the C1s (as seen in Fig 2 of the manuscript) which is not present in the shown C1s spectra. Moreover, the signal to noise ratios of O1s vs C1s seem to indicate that there is only a very low amount of Carbon present on the CO₂-treated sample, a no signs of Li₂CO₃ in the C1s that could explain the pretty significant signal in the O1s spectrum.

Adding to this, the XPS spectra in Fig 2 of the manuscript show a very strong Li₂CO₃-C2s-signal before annealing that vanishes after annealing. However, the O2s-signal the authors attribute to CO₃²⁻ are still almost 50% of the O1s. The authors need to quantify these signals to gain insight whether the amounts of assigned species in the O1s, C1s and Li1s match (of course taking into account the error of quantification). My impression from the spectra shown is that the signals attributed to Li₂CO₃ in the O1s and Li1s are due to different species since the amount of the according C1s signal is far too low. Also the “after annealing” temperature is not mentioned and I have difficulties connecting Fig 2c and 2d. The “after annealing” might be the 300°C, but it's not clear.

Figure 2 (d) is a very weak experimental evidence since oxygen signals in XPS can originate from so many different surface adsorbates I would suggest to be very careful with the assignment of O1s binding energies to certain chemical configurations without the accompanying signals from the other Elements (in this case esp. C1s).

mRIXS is not explained anywhere in the manuscript. It is a pretty complex method that needs quite a bit of explanation and I'm not sure the authors have an idea what they are talking about. Explain it and explain the data obtained via mRIXS or do not mention it.

The mRAS and XAS data presented look interesting, however I'm not sure whether the interpretation is correct, since it is not fully supported by the XPS data. My impression is that the physical processes involved specifically in mRAS may be more complex, making a simple fingerprinting approach with references misleading. There could be cross-excitations on the surface where energy is transferred between neighboring atoms. I'm not enough expert to judge whether the mRAS interpretation is correct, but if its findings are not fully supported by established methods like XPS the authors need to give more details on the physical processes.

Reviewer #3 (Remarks to the Author):

In this work, the formation of surface inactive layers on the garnet-type solid electrolyte is investigated using the novel techniques. The results clearly demonstrate the formation mechanism of the Li_2CO_3 and LiOH . However, the mechanism is already well known, and few new things are found in this work.

Comments

1) The removal and re-deposition of surface Li_2CO_3 and LiOH at elevated temperatures and/or UHV have been reported by Refs. 15, 36, and R1.

2) In Figure 4(a), a peak around 532 eV in O1s is attributed to carbonate. This peak is rather strongly observed in the spectra recorded under UHV (bottom). However, C1s XPS shows no (or very weak)

peak in Figure 4(b). This discrepancy might be explained by assuming that the LiOH layer is formed after annealing as well as Li₂CO₃. According to the databases of XPS, O1s peaks for metal hydroxides and metal carbonates appear around 531 eV. In addition, Li 1s for LiOH and Li₂CO₃ appears around 55 eV. Therefore, it is difficult to determine the surface species only from O1s and Li1s spectra. Furthermore, the MS data (Figure 2(e)) demonstrates that the base pressure of H₂O is higher than CO₂, which also shows the possibility of the re-deposition of LiOH. If the LiOH layer is thin (~ few nm), AP-mRAS might not clearly detect LiOH (Figure 3(c), (e)).

3) If LiOH is formed on the surface of annealed LLZO as indicated in the previous comment 2, the formation and/or growth of Li₂CO₃ layer (Figure 4(b)) is not the direct reaction of LLZO and CO₂, but the reaction of surface LiOH and CO₂.

4) In the Figure 4(d), a peak around 531 eV is assumed to be related to the Li extraction with references. Ref. 33 explains the O-K pre-edge peak is caused by the hybridization of transition metal d-orbital and oxygen 2p-orbital, and the peak shift is related to the change in the oxidation state of the transition metal. On the contrary, in this work, there is no element that changes its oxidation state, as it is assumed that lithium ions are extracted together with oxygen from the lattice.

Minor comments

5) The specimens should be described in more detail. For example, where were the LLZO pellets polished? In a glovebox or in air?

6) Composition of the solid electrolyte: which is true, Ta_{0.25} (in the abstract), or Ta_{0.5} (in the method section)?

7) Standard binding energy of C1s XPS: which is true, 284.6 eV (line 121), or 284.8 eV (line 373)?

Reference

[R1] Zhu et al., Adv. Energy Mater. 2019, 9, 1803440.

Point-to-Point Response Letter to Reviewers' Comments

Dear Reviewers,

The authors greatly appreciate your insightful comments and careful review on our manuscript. Please find enclosed our point-to-point response to reviewers' comments and revised manuscript, which we would like to submit as the revised version of NCOMMS-23-29214. This paper has been revised carefully according to the comments of the reviewers. Changes made to the manuscript have been highlighted in PDF file. The point-to-point responses to reviewers' comments are as following.

Reviewer #1 (Remarks to the Author):

This manuscript described a study using multiple X-ray spectroscopy method (in-situ) with different probing depths to study the solid electrolyte material. Theoretical understanding based on DFT was discussed. The reviewer found the manuscript at its current stage is not sufficient to be considered for publication at Nature Comms., primarily due to the below concerns. Major revision is required before potential reconsideration/re-submission.

Response: We greatly appreciate the positive comments and valuable suggestions from the reviewer.

Concern 1. Detailed characterization via multiple surface sensitive (in-situ) probes of garnet based solid electrolyte had already been very well documented and reaction mechanism/path was already understood. Noted the author adopted resonant Auger spectroscopy, which was not commonly used for garnet solid electrolyte surface. The rationale for using this methods is not clear and the reviewer did not find sufficient amount insights that was not previously discussed/reported.

Response: Thank you for your concern. Most previous reports have observed the final state of the reaction, and to our knowledge, there has never been an article that observed the thermodynamics and kinetics of the reaction *in situ* on a clean LLZO surface. Therefore, what they observed before is always the final product of the reaction rather than the entire process. This is also why, as mentioned in your comment below, "they observe LLZTO reactions with water most immediately while the reactions with CO₂ in dry conditions are very slow". Only some calculation results can show that the reaction of CO₂ is thermodynamically more preferential, but it has not been observed by the experiment. The initial interface layer generated on a clean electrolyte may have a certain impact on the air stability of the material, which cannot be observed in previous work. Only by utilizing more advanced characterization methods can we obtain the true reaction process. Near ambient pressure technology can slow down the rapid reaction process, which enable us to observe them. In a just published article *ACS Energy Lett.* 2023, 8, 3476–3484, the authors use GIXRD and APXPS to study the

surface reaction of LLZO, their results also support CO_2 can directly react with LLZO and is more active than H_2O . Therefore, we think that the reaction mechanism and pathway have not been already understood in previous study, and our results fundamentally reveal the new and actually reaction process.

The human factor in peak fitting has always been a problem in XPS data processing. This is also the driving force behind our desire to develop the mRAS and AEY method. The mRAS and AEY method can be implemented at any synchrotron radiation XPS end-station with almost no need to add new hardware. The mRAS has a detection depth slightly higher than XPS which not only can assist XPS in species identification, but also provide deep analysis abilities. In addition, its identification of species is also clearer and more intuitive because increasing dimensions can result in more fingerprint features just like mapping of resonant inelastic X-ray scattering (mRIXS). Due to the dimensional improvement, mRIXs has played an important role in the study of oxygen redox in cathode materials. We hope and have already promoted mRAS to the research of energy and catalytic materials (Angew. Chem.2023,135, e202215, *ACS Catal.* 2023, 13, 1, 11–18). In order to make our mRAS display clearer and more reliable, **we have added an *in situ* experiment on Li foil.** Li metal was *in situ* scraped using a Wobble-Stick with sharp blade surface. By utilizing near ambient pressure technology, we can obtain pure Li_2O , LiOH and Li_2CO_3 . We have added the standard results of these references to Fig. 4, as shown in the following figure. Through their comparison, we can have a clearer understanding of the types and changes of products. Using mRAS, it can be clearly seen that LLZO after annealing treatment does not have Li_2O on the surface.

Fig. 4 | The mRAS and XAS spectra of pure LiOH, Li_2O , Li_2CO_3 and LLZO surface at different states.

(a) The color-coded mRAS spectra comparison of LLZO with pure LiOH, Li_2O ,

Li₂CO₃. (b) The color-coded mRAS spectra comparison of LLZO surface at different states, corresponding to the process (a)(e)(f)(h) in flowchart Fig. 2. (c) Distribution of LLZO surface at different states in photon energy direction. (d) Distribution of LLZO surface at different states in KE direction. LLZO (blue line) is extracted from $h\nu=531.9$ eV, Li₂CO₃ (red line) is extracted from $h\nu=533.7$ eV and LiOH (green line) is extracted from $h\nu=532.8$ eV.

Another important reason for the rational of using it is that solid electrolytes are very poor electronic conductors, severe charging effects exist during XPS testing at room temperature. The authors of above *ACS Energy Lett* paper mentioned that “significant asymmetric broadening and shifting of peaks and variations in intensities were observed for all samples due to charging, especially at low temperatures, complicating analysis.” Compared with XPS, the photon energy of the peak is not influenced by the charging effect in mRAS and AEY. Changes in kinetic energy caused by charging can be corrected using specific fingerprint characteristics in mRAS. Therefore, we can clearly identify the various reaction products that are difficult to provide in their article. Thus, mRAS and AEY can play an important role when XPS analysis encounters difficulties.

Concern 2. the reviewer found the discussion remained shallow with limited new understanding on the reaction mechanism and steps.

Response: Thank you for your concern. We added a flowchart as Fig.2 in the manuscript and try our best to explain each reaction step and mechanism clearly. The main processes revealed in our manuscript are as follows: A very thin layer of Li₂CO₃ will be formed on polished LLZO pellet after exposed to air for 15 minutes. Vacuum annealing at 350 °C can remove this layer through the reaction $x\text{Li}_2\text{CO}_3 + \text{Li}_{6.5-2x}\text{H}_2\text{La}_3\text{Zr}_{1.5}\text{Ta}_{0.5}\text{O}_{12} \rightarrow \text{Li}_{6.5}\text{La}_3\text{Zr}_{1.5}\text{Ta}_{0.5}\text{O}_{12} + x\text{H}_2\text{O} + x\text{CO}_2$. However, surface contaminate C cannot be removed. After annealing with 1×10^{-6} mbar O₂, the surface carbon will significantly decrease. The clean LLZO surface is unstable, and a very small amount of lithium carbonate forms on the surface during the cooling process under UHV.

LLZO can directly react with CO₂ even at a very low pressure $\sim 10^{-8}$ mbar and near equilibrium at 10^{-2} mbar. H₂O has a higher reaction pressure $\sim 10^{-2}$ -0.5 mbar with more reaction products. The reaction of CO₂ with LLZO to form Li₂CO₃ is thermodynamically favored due to its lower absorption energy compared to the reaction with H₂O at low pressure. CO₂ reaction is restricted by the hindered oxygen supply from the sub-layer. The Li⁺/H⁺ exchange is more intense when sufficient H₂O is adsorbed on the LLZO surface, leading to more reaction products. We have strengthened the description of the mechanism and steps in the manuscript, explaining the entire process in an organized manner.

Fig. 2 | Schematic diagram of the entire *in situ* experimental process.

(a-b). Polished LLZO pellet was vacuum annealing at 350 °C for 30 min to remove surface Li_2CO_3 , but surface contaminated C cannot be removed. (c) During the cooling process, a small amount of Li_2CO_3 will be generated on the surface of LLZO. (d-e) Most surface contaminated C can be removed by annealing in 1×10^{-6} mbar O_2 at 350 °C for 30 min. (f-h) The clean LLZO surface then be used to investigate the reaction with CO_2 and H_2O .

To strengthen the discussion section, we accurately measured the XPS peak positions and half width corresponding to these species, as shown in the following figure. Thus, we can fix Li_2O peak position at 528 eV with a FWHM of 1.35 eV; LiOH at 530.9 eV with a FWHM of 1.61 eV, Li_2CO_3 at 531.5 eV with a FWHM of 1.75 eV. These data make our peak fitting more accurate. By utilizing them, we optimized the peak fitting results and calculated changes in components during the reaction process.

Fig.S4. APXPS results of Li metal in different gases.

With precise peak fitting, we can conduct quantitative analysis of the products during the reaction process. The results of product changes during CO_2 and H_2O reactions are shown in the following figure and table. We calculated the atomic ratio of O/C using

the photoionization cross-sections of O (0.3383) and C (0.1308) at 650 eV. From the table, we can see that the O/C is maintained at a stable value ~ 3.9 during the pressure range 10^{-6} - 10^{-2} mbar. **The stability of the ratio means that within this pressure range, the surface is relatively pure Li_2CO_3 without LiOH and the layer is very thin, and the X-ray can detect all information within the layer.** Compared with pure Li_2CO_3 ~ 2.48 , the ratio is higher. This phenomenon can be explained simply by the following figure. When 650 eV X-ray is incident on the surface of the material, the photoelectron kinetic energy of O is about 120 eV, while that of C is 370 eV. The detected depth of C is larger, so for pure Li_2CO_3 , the proportion of C will be higher. For the results above 0.5 mbar CO_2 , O/C is affected by the gas peak, the higher the CO_2 pressure, the lower the ratio. Interestingly, after we pumped 1 mbar CO_2 back to the vacuum, the O/C ratio slightly decrease to 3.75, indicating that the thickness of the Li_2CO_3 layer slightly exceeded the detection depth of O photoelectrons.

Fig. S6. The O/C ratio changes during the CO_2 reaction which is used to demonstrate surface reaction products. A simple explanation of the phenomenon.

For the reaction of H_2O , we can clearly see the changes in the content of Li_2CO_3 and LiOH during the reaction process. Li_2CO_3 will rapidly increase under low pressure because our gas path can only be cleaned up to 1×10^{-7} mbar, and there will still be a small amount of CO_2 present in the gas path. This also indirectly indicates that CO_2 is more likely to directly react with LLZO than water under low pressure.

We added these discussions to the manuscript and included these data and figures in the SI.

Despite the concerns, the reviewer do find the merits of the proposed experimental flow. The reviewer recommend that this methods/experimental flow could be quite powerful only if applied to different solid electrolyte surface/interface systems for a comparative study. New insight could be delivered from difference associated with chemistry, composition, micro-structure and etc. It is a bit unfortunate that those comparison are not included in this manuscript.

Response: Thank you for your positive advice. We added *in situ* ambient pressure experiment on $\text{Li}_{1.5}\text{Al}_{0.5}\text{Ge}_{1.5}\text{P}_3\text{O}_{12}$ (LAGP), which is an air stable solid electrolyte. The LAGP ceramic pellet was removed from its sealed packaging and placed directly into the test chamber from air without polishing, which is labelled as “pristine”. The signals of P, Ge and O can still be seen on the surface, indicating that LAGP has much better air stability than LLZO. Vacuum annealing at 350 °C hardly changes the composition of the surface. Therefore, we annealed the sample in 1×10^{-6} mbar O_2 at 350 °C for 30 minutes. As shown in the figure below, it can be seen that the surface C of the sample at 350 °C significantly decreases, while the signals of O and P are significantly enhanced. However, when the sample was cooled to room temperature, the signal of C increased significantly while the signal of O and P decreased. No such carbonization phenomenon was observed on the surface of LLZO.

Fig. S17. Evolution of surface species on LAGP during annealing and cooling processes.

We conducted APXPS on LAGP sample and the results are shown in the following figure. LAGP cannot react with CO_2 and H_2O even at a high pressure of 0.5 mbar. By comparing two electrolyte materials, we infer that the stability of LAGP comes from the hydrophobic carbon layer generated on the surface and the stability of PO_4 structure. Due to the stability of surface O, it is difficult for LAGP to directly react with CO_2 to generate Li_2CO_3 . On the contrary, LLZO surface can react directly with CO_2 to form Li_2CO_3 , which is a hydrophilic layer, may cause the occurrence of subsequent severe reactions. A similar phenomenon was also observed in a just published article comparing the air stability of $\text{NaNi}_{1/3}\text{Fe}_{1/3}\text{Mn}_{1/3}\text{O}_2$ (NFM) and $\text{Na}_3\text{V}_2(\text{PO}_4)_3$ (NVP). (*Advanced Functional Materials* (2023): 2308257). The results indicate that the hydrophilicity and hydrophobicity of the initial products on the clean electrolyte surface, as well as the stability of the surface O structure, may play an important role in determining the air stability of solid electrolyte materials.

We have added these comparisons in our manuscript in the “Comparison of LLZO with LAGP” part and added these two figures in Supplementary as Fig. 17 and Fig. 18.

.

Fig.18. *In situ* ambient pressure experiments on LAGP sample with CO₂ and H₂O.

Some additional technical comments are provided below.

Fig 3d. the polished LLZO surface seemed fully covered Li₂CO₃ with very weak signal of Ta. Was the polishing properly done? Or the sample got exposed to air/moisture/CO₂ during sample transfer? A properly polished LLZO should be free of Li₂CO₃ and at least show strong La, Ta, Zr signals.

Response: Thank you for your comment, the polishing process is properly done, the LLZO pellet was polished in air by sandpapers with grit number from 800 to 1500 to achieve parallel faces. The polishing thickness was sufficient to ensure that the surface contamination layer is completely polished off and then the surface was wiped by alcohol. The polished pellet was mounted onto a sample holder under ambient air (which took about 15 min) and were pumped into the instrument. **We have added these experiment details into the methods section.**

LLZO quickly forms a layer of Li₂CO₃ on its surface after being exposed to air, the detection depth of XPS and mRAS is very shallow (1-3 nm), so it will display a strong Li₂CO₃ signal. We have tried polishing the pellet in the glove box and transferring the sample using a glove bag, but still cannot prevent the formation of Li₂CO₃. Therefore, annealing is an indispensable step to obtain clean LLZO surface. From the XPS survey

spectra shown in the figure below, the peaks of La and Zr are much more obvious after vacuum annealing. We have added an explanation as “The polished LLZO pellet was mounted onto a sample holder under ambient air (which took about 15 min) and were pumped into the instrument. Thus, a strong signal of Li_2CO_3 and weak signals of La and Zr could be observed in XPS spectra as shown in Fig. 3(c) and Supplementary Fig. 1” in the manuscript.

Fig. S1. After vacuum annealing, the signals of La and Zr on the surface of LLZO are significantly enhanced.

It is very misleading discussion on the point of oxygen supply at sub-layer when the authors discuss CO_2 reaction with LLZTO garnet surface. First, the LLZTO + dry CO_2 reaction, if this could kinetically occur, is a solid-gas reaction. There is no available "ions" that can possibly participate. It is only Li and O atoms sitting on the crystal-graphic site on the surface and near the surface that reacts with CO_2 . Therefore the reaction formula $2\text{Li}^+ + \text{O}^{2-} + \text{CO}_2 \rightarrow \text{Li}_2\text{CO}_3$ is very misleading and author should consider use proper defect chemistry note to rewrite the reaction. Third, LLZTO is not a good oxygen ionic conductor at least at the temperature of this study and mostly "oxygen ion" moves via oxygen vacancies. The fact that dry CO_2 reaction with LLZTO to form Li_2CO_3 has nothing to do with so called migration of oxygen ion from the sublayer to surface. This physical image is not realistic and would entail crystal structure and micro-structure change, based on which experimental results, the author rule out the possibility of CO_2 react with Li_2O at the surface of LLZTO?

Response: Thank you for your comments. The reaction formula $2\text{Li}^+ + \text{O}^{2-} + \text{CO}_2 \rightarrow \text{Li}_2\text{CO}_3$ is indeed very misleading. We have removed all ion descriptions from the manuscript and changed the reaction formula to “ $\text{Li}_{6.5}\text{La}_3\text{Zr}_{1.5}\text{Ta}_{0.5}\text{O}_{12} + x\text{CO}_2 \rightarrow \text{Li}_{6.5-2x}\text{La}_3\text{Zr}_{1.5}\text{Ta}_{0.5}\text{O}_{12-x} + x\text{Li}_2\text{CO}_3$ ”. In the manuscript, we also believe that “LLZTO is not a good oxygen ionic conductor at room temperature and bulk O cannot move to the surface. Thus, the reaction products of CO_2 are very little.” We have changed the description in the manuscript to “In contrast, LLZO is not a good oxygen ionic conductor at room temperature, the oxygen in the sub-layer have difficulty

migrating to the surface”.

The peak position of O 1s between LLZO and Li₂O is over 1 eV, and the difference is more pronounced on mRAS. No Li₂O was observed on the surface of the annealed sample by XPS and mRAS, so the influence of Li₂O can be completely ruled out. We added the following figure in Supplementary Fig. 4, and added a description as “No signal of Li₂O was detected in the O 1s XPS and O K edge AEY spectra of annealed LLZO as shown in Supplementary Fig. 4” in the manuscript.

Fig. S4. Comparison of annealed LLZO and Li₂O to exclude the generation of Li₂O during the annealing process.

That theoretical explanation of adsorption capacity is quite misleading too. First, what is the adsorption capacity, mole of CO₂/H₂O molecular per LLZTO surface area? Second, how is this adsorption capacity (semi)-/quantified such that the authors can possible compare that capacity is high or low? Third, how is the capacity (mostly likely a thermodynamic term) determines the reaction rate (a kinetic term?) Lastly, there exists numerous report that LLZTO reacts with water almost immediately while the reactions with CO₂ in dry conditions are very slow. How does that reconcile author's statement that CO₂ is more active than H₂O on LLZTO surfaces. Could the author extract based on the very rich experimental observation a reason rate of constant to support the discussion?

Response: Thank you for your comments. Indeed, the use of “adsorption capacity” in the manuscript is inappropriate, and we have changed it to “adsorption energy”.

In our theoretical explanation, we think the surface adsorption energy determines the initial reaction pressure. We calculated that the adsorption energy E_{ab} of CO₂ on the surface of LLZO is about -6 eV, while the E_{ab} of H₂O is much higher \sim -1 eV. Thus, the calculated results support that the reaction of CO₂ with LLZO to form Li₂CO₃ is **thermodynamically favored** as observed by our experiment. Here, we use the different E_{ab} to explain why CO₂ preferentially reacts with LLZO at low atmospheric pressure.

For kinetic processes, if we consider the energy barrier or intermediate process, the calculation is very complex and difficult to be completed. Thus, we chose to calculate the exchange energy E_{ex} of the reaction with H_2O to prove that Li^+/H^+ exchange is very easy to occur after H_2O adsorption. The results are used to support the reaction product of H_2O is much larger than CO_2 at high pressure.

It is very easy to explain the inconsistent to previous reports as the response of concern 1. In the previous report, they always observe the final state of CO_2 reaction. When they started observing, the reaction of CO_2 with LLZO had already ended, thus, nearly no changes can be observed. Of course, they cannot determine whether CO_2 can directly react with LLZO. In the Supplementary Fig. 10 and TOC of the manuscript, we plotted the trend of the amount of reaction products as a function of pressure.

The reviewer found Li gradient as result of reaction with CO_2 is not well supported. The author claimed the system may have an oxygen stoic difference. This itself is not supporting Li gradient since LLZTO charge neutrality by change in oxygen stoic does not have to be compensated by Li stoichiometry, especially when the off stoic variation is very small value. Additional experimental proof is needed to confirm the gradient of Li.

Response: In the manuscript, our viewpoint is that Li can move to the surface to react with CO_2 , while the internal O content remains unchanged, leading to an increase in the oxygen valence state. By observing the variation of the O valence state with depth, we can infer how much Li has been removed. This is like using changes in transition metal valence states in cathode materials to calibrate lithium removal. Because the reaction products of CO_2 are very single (only Li_2CO_3) and the valence states of La, Zr, and Ta in LLZO are not easily changed. Therefore, we believe that it is reliable to obtain the Li gradient through changes in the valence states of O in AEY and TEY spectra. It has also been reported that an increase in the valence state of O can cause the absorption spectrum peak position to shift towards the lower energy direction, as shown in the following figure *PLoS one* 7.11 (2012): e49182. We also attempted to use *in situ* chips for transmission electron microscope. However, due to the high requirements for sample handling and atmosphere, we didn't get the necessary data. This also

demonstrates the uniqueness and advantages of our experimental method.

Reviewer #2 (Remarks to the Author):

The authors present an interesting paper and try to shed some light on a relevant question in the field of Garnet electrolyte research. They employ some uncommon experimental methods, which promises new insights not possible before.

However, some of the methods employed involve a quite complex physics and may be misleading. The authors should give more details on some of those experiments, especially where the conclusions drawn are unexpected or contradicting to other findings.

Response: We greatly appreciate the positive comments and valuable suggestions from the reviewer.

The abstract already has some strange flaws, raising some scepticism:

Line 22: „The air stability of solid state electrolyte should be overcome for successful commercialization of all-solid-state Li-ion battery.” This doesn’t make sense.

Response: Thank you for your suggestion. We changed the sentence to “Garnet-type $\text{Li}_{6.5}\text{La}_3\text{Zr}_{1.5}\text{Ta}_{0.5}\text{O}_{12}$ (designated as LLZO) is considered a promising solid electrolyte because of its high Li^+ conductivity and superior stability against metallic Li. However, the surface degradation of LLZO in air hinders its application for all-solid-state battery and the mechanism is still unclear.”

Line 23: “However, the surface degradation mechanism of solid state electrolyte is still ambiguous due to the lack of powerful in situ gas/solid interface characterization tools.” A surface degradation mechanism cannot be ambiguous.

Response: Thank you for your suggestion, we changed “ambiguous” to “unclear”.

Line 48: “recent findings” with a citation from 2014. Well, that’s not very recent.

Response: Thank you for your suggestion. We changed the “recent” to “previous”.

There is a recent report from the Grey group, <https://pubs.acs.org/doi/full/10.1021/acsenergylett.3c01042>, that deals with the same questions and uses some overlapping methods. The authors need to consider this paper in their arguments. Specifically, in that paper the change in the O1s at around 300 °C is linked to protonation, while no change in Lithiumcarbonate can be found. Their interpretation is much better supported by experimental evidence imo.

Response: Thank you very much for your suggestion. The article had not been published when we submitted our manuscript, so we did not consider this paper in our arguments before. I have carefully read the article, Grey group indeed did a very detailed and systematic work. Their results are quite consistent with our conclusion, that is, we both observed that CO_2 is more likely to react directly with LLZO than H_2O . **We have considered this paper in our arguments as reference 38.** Comparing with

their results, we can demonstrate some advantages of our characterization as following:

1. The LLZO surface we processed is much cleaner than theirs, which is important for studying the surface reaction mechanism.
2. In that paper, the authors mentioned that “Significant asymmetric broadening and shifting of peaks and variations in intensities were observed for all samples due to charging, especially at low temperatures, complicating analysis.” Compared with XPS, our XAS data is not influenced by the charging effect. It can play an important role when XPS analysis encounters difficulties.
3. In that paper, the authors cannot give the clear identification of surface oxygen such as Li_2CO_3 and LiOH , and even the assignment of LLZO may be influenced by Li_2O . Our mRAS method can give clear identification of these species. In addition, we also provide standard XPS, XAS and mRAS spectra of Li_2O , LiOH and Li_2CO_3 for species identification and analysis. These will be presented in detail when answering the following comments.

In that paper, O 1s at around 300 °C was measured in H_2O environment, which is different from the conditions of vacuum annealing. For the vacuum annealing of LLZO, there are some literatures that can support our results (Refs. 15, 36, ACS Appl. Energy Mater. 2018, 1, 7244–7252. Adv. Energy Mater. 2019, 9, 1803440.) We prepared LLZO with a clean surface after multiple explorations based on their annealing conditions. The following figure shows a comparison of our O 1s XPS results with literature Adv. Energy Mater. 2019, 9, 1803440. The authors of the AEM said that “the annealed LLZO data represent, to our knowledge, the only reported XPS analysis of LLZO without any interfacial surface reaction species.” The difference in annealing temperature may be caused by different positions of thermocouples, as our thermocouples are directly pressed onto the surface of pellet. In addition, almost all of our XPS data are supported by simultaneous measurement of AEY and TEY, and their results are highly consistent with XPS results. Thus, the decomposition of lithium carbonate at around 300 °C through the reaction $x\text{Li}_2\text{CO}_3 + \text{Li}_{6.5-2x}\text{H}_{2x}\text{La}_3\text{Zr}_{1.5}\text{Ta}_{0.5}\text{O}_{12} \rightarrow \text{Li}_{6.5}\text{La}_3\text{Zr}_{1.5}\text{Ta}_{0.5}\text{O}_{12} + x\text{H}_2\text{O} + x\text{CO}_2$ is reliable and our XPS and MS results in Fig.3(d-e) clearly reveal the reaction mechanism.

result in Adv. Energy Mater.
2019, 9, 1803440

our result

O 1s XPS of LLZO

The partial pressure dependent measurements are new and interesting! But the description is lacking some experimental details. How where the gas flows controlled, how where the partial pressures determined?

Response: Thank you for your suggestion. We have added the experimental details in the Methods part as “High-purity CO₂ and water vapor were introduced into the reaction chamber by controlling two independent all-metal leak valves (VACGEN) as shown in Fig.3(a). The gas pressure was read by a capacitance film vacuum gauge (PFEIFFER CMR 363) attached on the chamber. All spectra were collected after the pressure had stabilized for 15 min. For the experiments of a mixture of H₂O + CO₂, the water vapor of 0.5 mbar was firstly maintained, then CO₂ was introduced until the total pressure stabilized at 1 mbar.”

The authors also need to give more experimental details with regard to sample preparation and handling, since this is so crucial in the case of LLZO. How where the samples transferred between experiments? Wenn where samples broken to create fresh surface? How where the surfaces prepared, polished, etc.?

Response: Thank you for your suggestion. We have added the details of sample preparation in the method section (red color) as “The starting materials of Li₂CO₃ (Alfa Aesar, 99.9%), La₂O₃ (Alfa Aesar, 99.9%), ZrO₂ (Alfa Aesar, 99.5%) and Ta₂O₅ (Aladdin, 99.5%) were mixed in stoichiometric amounts with 15 mol% Li₂CO₃ in excess. The mixture was ball-milled in 2-propanol for 12 h with agate balls in an agate vial, and then dried and heated in air at 1150 °C for 12 h. Then the ball-milling was repeated once, and the powder was sieved with a mesh number of 600 to obtain fine

particles. The pellets were made by hot-pressing of the as-prepared LLZO powder in a flowing argon atmosphere at a temperature of 1050 °C under a constant pressure of 50 MPa for 1 hour.”

We have added the experimental details in the methods section (highlighted) as “The samples were brought to the SSRF through an aluminum plastic bag sealed in a glove box. Before transferred to ultra-high vacuum chamber, the LLZO pellet was polished in air by sandpapers with grit number from 800 to 1500 to achieve parallel faces. The polishing thickness was sufficient to ensure that the surface contamination layer is completely polished off and then the surface was wiped by alcohol. The polished pellet was mounted onto a sample holder under ambient air (which took about 15 min) and were pumped into the instrument.”

The authors claim that the XPS-spectra in Fig 1 of the supplementary show signs of Li_2CO_3 after annealing. However, lithium carbonate shows a very distinguished signal in the C1s (as seen in Fig 2 of the manuscript) which is not present in the shown C1s spectra. Moreover, the signal to noise ratios of O1s vs C1s seem to indicate that there is only a very low amount of Carbon present on the CO_2 treated sample, a no signs of Li_2CO_3 in the C1s that could explain the pretty significant signal in the O1s spectrum.

Response: Thank you for your comment. We are sorry that our description of the experimental process was not particularly clear, so we have added a flowchart in the manuscript as Fig 2 to make the entire process more clear. The flowchart is shown in the following figure. **The C 1s in Fig. 2 (now Fig. 3) of the manuscript is measured at room temperature corresponding to process (c) in the following figure. While the spectra in Supplementary Fig 1 (now Fig.3) is measured at 350 °C, corresponding to process (b).** At 350 °C, there is no Li_2CO_3 on the surface of LLZO, thus only the peak of lattice O and surface contaminated C can be observed in supplementary Fig 1 (now Fig.3). The thin Li_2CO_3 is formed during the cooling process. We added “after vacuum annealing at room temperature” in Fig. 2(c) (now Fig.3(c)).

Fig. 2 | Schematic diagram of the entire *in situ* experimental process.

(a-b). Polished LLZO pellet was vacuum annealing at 350 °C for 30 min to remove

surface Li_2CO_3 , but surface contaminated C cannot be removed. (c) During the cooling process, a small amount of Li_2CO_3 will be generated on the surface of LLZO. (d-e) Most surface contaminated C can be removed by annealing in 1×10^{-6} mbar O_2 at 350°C for 30 min. (f-h) The clean LLZO surface then be used to investigate the reaction with CO_2 and H_2O

As shown in the flowchart process (d-e), before CO_2 treatment, we annealed the sample in 1×10^{-6} mbar O_2 at 350°C for 30 minutes. The purpose of this step is to reduce surface contaminated carbon and prevent their strong signal from affecting our observation of carbonate changes in C 1s XPS spectra. From the comparison of process b and d spectra in the following figure, it can be seen that after O_2 annealing, the signal of surface contaminated carbon is significantly reduced, while the signal of all other elements is improved, indicating that more LLZO signals are displayed. Supplementary Fig.1(c-d) (now Fig.9(a-b)) are corresponding to process (g), at 350°C , the Li_2CO_3 cannot be totally removed by vacuum annealing as the H in LLZO is fully removed. Therefore, there will be weak lithium carbonate signals on O 1s even at high temperatures. Surface contaminated C has undergone O_2 annealing and secondary annealing, thus the signal is much weaker, which show a poor signal-to-noise ratio compared with O 1s peak. To prevent misunderstandings, we have split the figures in Original Fig. S1 into two figures as supplementary Fig. 3 and Fig. 9.

Figure. Survey of LLZO after vacuum annealing and 1×10^{-6} mbar O_2 annealing.

Adding to this, the XPS spectra in Fig 2 of the manuscript show a very strong Li_2CO_3 -C2s-signal before annealing that vanishes after annealing. However, the O2s-signal the authors attribute to CO_3^{2-} are still almost 50% of the O1s. The authors need to quantify these signals to gain insight whether the amounts of assigned species in the O1s, C1s and Li1s match (of course taking into account the error of quantification). My impression from the spectra shown is that the signals attributed to Li_2CO_3 in the O1s and Li1s are

due to different species since the amount of the according C 1s signal is far too low. Also the “after annealing” temperature is not mentioned and I have difficulties connecting Fig 2c and 2d. The “after annealing” might be the 300°C, but it’s not clear.

Response: Thank you for your comment. The “after annealing” temperature in Fig.3(c) is room temperature, while in Fig.S3 and Fig.S9 are 350 °C. We added “at RT” in Fig.3(c).

The most important reason for the mismatch between O and C peak intensities in Fig 2 (now as Fig 3) is that **the comparison signals come from different species**. The peak of surface contaminated carbon is so strong that it shows weak C 1s signals of carbonate in Fig.3(c). **After O₂ annealing, the signal of surface contaminated C is significantly reduced, then we can see a clear peak of carbonate as shown in the following figure.**

The intensity difference between O 1s peak and C 1s peak also be affected by the atomic ratios and photoionization cross-sections: for Li₂CO₃ on the LLZO surface, the atomic ratios of oxygen and carbon are 3:1 and have different photoionization cross-sections (O is 0.3383 and C is 0.1308 at 650 eV). **The calculated O/C area ratio for Li₂CO₃ is approximately 7.76:1 at 650 eV, so it appears that the peak of O is more pronounced than that of C.**

We found that O 1s and C 1s peaks at different stages during annealing and CO₂ experiment can be well fitted with a Gaussian peak corresponding to CO₃²⁻. Thus, we quantified the changes of O and C during annealing and CO₂ reaction processes. The results are shown in the following figure. The main reason for not quantifying Li 1s is that the signal of lithium itself is weaker and have a large freedom in peak fitting. We calculated the atomic ratio of O/C using the photoionization cross-sections of O (0.3383) and C (0.1308) at 650 eV. From the table, we can see that the O/C is maintained at a stable value ~3.9 during the CO₂ pressure range 10⁻⁶-10⁻² mbar. **The stability of the ratio means that within this pressure range, the surface is relatively pure Li₂CO₃ without LiOH and the layer is very thin, and X-ray can detect all information within the layer.** Compared with pure Li₂CO₃ ~2.48, the ratio is higher. This phenomenon can be explained simply by the following figure. When 650 eV X-ray is incident on the surface of the material, the photoelectron kinetic energy of O is about 120 eV, while that of C is 370 eV. The detected depth of C is larger, so for pure Li₂CO₃, the proportion of C will be higher.

For the sample after vacuum and O₂ annealing, the ratio seems higher ~4.5 than Li₂CO₃ ~3.9, it may be caused by the weak signal of Li₂CO₃ in C 1s resulting in some fitting error. For the results above 0.5 mbar CO₂, O/C is affected by the gas peak, the higher the CO₂ pressure, the lower the ratio. Interestingly, after we pumped 1 mbar CO₂ back to the vacuum, the O/C ratio slightly decrease to 3.75, indicating that the thickness of the Li₂CO₃ layer exceeded the detection depth of O photoelectrons. **We have added some explanation in the manuscript and added the following figure in the supporting information as Fig. 6.**

Fig. S6. The O/C ratio changes during the CO₂ reaction which is used to demonstrate surface reaction products. A simple explanation of the phenomenon.

Figure 2 (d) is a very weak experimental evidence since oxygen signals in XPS can originate from so many different surface adsorbates I would suggest to be very careful with the assignment of O 1s binding energies to certain chemical configurations without the accompanying signals from the other Elements (in this case esp. C 1s).

Response: Thank you for your comment. Within the detection depth of XPS, although we cannot guarantee that the surface species are only Li₂CO₃, the O 1s and C 1s peak can be well fitted using a Gaussian peak of Li₂CO₃. We also added an *in situ* experiment specifically to determine the peak position and half width of Li₂CO₃. From the figure below, we can determine that the peak position of Li₂CO₃ is 531.5 eV, with a half width

at half height of 1.75. This is very close to the fitting results of the O 1s changes during the annealing process. C 1s only contains Li_2CO_3 , the peak fitting will be more accurate. The results indicate that most of the surface species is Li_2CO_3 , and the amount of other species are small, which can not affect our observation of the tendency. In Figure 2(d) (now Figure 3(d)), the orange line represents the changes in O 1s, **while the blue line represents the change in C 1s**, which are obtained through peak fitting. Their sudden changes temperature $\sim 300^\circ\text{C}$ exhibit a high degree of consistency. Therefore, we believe that the results are very reliable, which is further confirmed by the results of mass spectrum. Both CO_2 and H_2O were simultaneously observed as the decomposition products around 300°C . Thus, the actual reaction at the LLZO surface during the vacuum annealing process could be: $x\text{Li}_2\text{CO}_3 + \text{Li}_{6.5-2x}\text{H}_{2x}\text{La}_3\text{Zr}_{1.5}\text{Ta}_{0.5}\text{O}_{12} \rightarrow \text{Li}_{6.5}\text{La}_3\text{Zr}_{1.5}\text{Ta}_{0.5}\text{O}_{12} + x\text{H}_2\text{O} + x\text{CO}_2$.

mRIXS is not explained anywhere in the manuscript. It is a pretty complex method that needs quite a bit of explanation and I'm not sure the authors have an idea what they are talking about. Explain it and explain the data obtained via mRIXS or do not mention it.

Response: Thank you for your suggestion. We provide a simple explanation of the mRAS process and mRIX process in Fig.S5 as the following figure. From the figure, it can be seen that mRAS and mRIXs have many similarities: one detects emitted electrons and the other detects emitted photons around the absorption threshold. For soft X-ray, the photon yield is one percent of the electron yield and the efficiency of electronic energy analyzers is often much stronger than that of grating spectrometers, which means that mRAS has much lower requirements for photon flux and acquisition time than mRIXs. In addition, the mRAS and AEY method can be implemented at any synchrotron radiation XPS end-station with almost no need to add new hardware. Due to the dimensional improvement, mRIXs has played an important role in the study of oxygen redox in cathode materials. Therefore, we mention mRIXs here in the hope of showcasing to readers the advantages of our experimental method and potential functions similar to mRIXs in related fields.

We do not have mRIX data in the manuscript, so we removed it from the main text and placed it in SI as following.

Fig. S5. Schematic diagram of the fundamental process of normal Auger and resonant Auger. Compared to normal Auger, resonance Auger contains valence band information of the sample. The core level electron can be excited under irradiation, followed by two parallel de-excitation channels (photon-in-electron-out vs. photon-in-photon-out), where the energy distribution of emitted electrons and photons can be further resolved into mRAS and mRIXS. The two-dimensional maps provide energy resolution along both incident photon energy and emitted electron/photon axis, which can well disentangle the spectra overlapping effect. For soft X-ray, the photon yield is one percent of the electron yield and the efficiency of electronic energy analyzers is often much stronger than that of grating spectrometers, which means that mRAS has much lower requirements for photon flux and acquisition time than mRIXS. Using light from a bending magnet beamline, we can also complete a mRAS mapping in steps of 0.1 eV in 15 minutes. In addition, the mRAS and AEY method can be implemented at any synchrotron radiation XPS end-station with almost no need to add new hardware.

The mRAS and XAS data presented look interesting, however I'm not sure whether the interpretation is correct, since it is not fully supported by the XPS data. My impression is that the physical processes involved specifically in mRAS may be more complex, making a simple fingerprinting approach with references misleading. There could be cross-excitations on the surface where energy is transferred between neighboring atoms. I'm not enough expert to judge whether the mRAS interpretation is correct, but if its findings are not fully supported by established methods like XPS the authors need to give more details on the physical processes.

Response: Thank you for your comment. To make our species identification more accurate, we added mRAS of pure Li₂O, LiOH and Li₂CO₃ in Fig.4. By comparison, we can clearly identify various species.

Fig. 4 | The mRAS and XAS spectra of pure LiOH, Li₂O, Li₂CO₃ and LLZO surface at different states.

(a) The color-coded mRAS spectra comparison of LLZO with pure LiOH, Li₂O, Li₂CO₃. (b) The color-coded mRAS spectra comparison of LLZO surface at different states, corresponding to the process (a)(e)(f)(h) in flowchart Fig. 2. (c) Distribution of LLZO surface at different states in photon energy direction. (d) Distribution of LLZO surface at different states in KE direction. LLZO (blue line) is extracted from $h\nu=531.9$ eV, Li₂CO₃ (red line) is extracted from $h\nu=533.7$ eV and LiOH (green line) is extracted from $h\nu=532.8$ eV.

Our mRAS and XAS results are very consistent with our XPS results and have greatly assisted in our XPS data analysis. For XAS, taking the reaction process of H₂O on LLZO as an example, using our new differential method, we can obtain an evolution that is completely consistent with XPS. As can be seen in the following figure, for 1×10^{-4} mbar H₂O, the peak of Li₂CO₃ appeared in the yellow line. When the surface LiOH increases in XPS, it appears a significant peak of LiOH on the differential absorption spectrum, LLZO becomes sunken as the surface content decreases. With the introduction of 0.5 mbar H₂O+0.5 mbar CO₂, the peak of LiOH on XAS weakens and the peak of Li₂CO₃ becomes stronger. The changes observed by TEY are not as strong as those observed by AEY because TEY has a deeper detection depth.

For mRAS, as shown in the following figure, Li₂CO₃ appears clearly at 1×10^{-4} mbar H₂O, while LiOH basically covers the signals of other species at a H₂O pressure of 0.5 mbar. These results are very consistency with XPS results.

In response to the previous comment, we have shown a simple physical explanation of mRAS. We acknowledge that the physical processes involved are more complex. At current stage, we cannot provide accurate explanations for some phenomena such as when the O 2p orbital hybridize with an unfilled metal d orbital, that the spot will be twisted and localized. We are trying to explore and combine calculations to explain these phenomena in simpler systems such as copper oxidation in another preparing manuscript.

Reviewer #3 (Remarks to the Author):

In this work, the formation of surface inactive layers on the garnet-type solid electrolyte is investigated using the novel techniques. The results clearly demonstrate the formation mechanism of the Li_2CO_3 and LiOH . However, the mechanism is already well known, and few new things are found in this work.

Response: Thank you very much for your affirmation of our new techniques and experimental results. Most previous reports have observed the final state of the reaction, and to our knowledge, there has never been an article that observed the thermodynamics and kinetics of the reaction *in situ* on a clean LLZO surface. Therefore, what they observed before is always the final product of the reaction rather than the entire process. Only some calculation results can show that the reaction of CO_2 is thermodynamically more preferential, but it has not been observed by the experiment. Only by utilizing more advanced characterization methods can we obtain the true reaction process. Near ambient pressure technology can slow down the rapid reaction process, which enable us to observe them.

In addition, the previously unobserved initial interface layer generated on a clean electrolyte may have a certain impact on the air stability of the material. **We added *in situ* ambient pressure experiment on $\text{Li}_{1.5}\text{Al}_{0.5}\text{Ge}_{1.5}\text{P}_3\text{O}_{12}$ (LAGP) to explain the possible role of this initial product layer.** The LAGP ceramic pellet was removed from its sealed packaging and placed directly into the test chamber from air without polishing, which is labelled as “pristine”. The signals of P, Ge and O can still be seen on the surface, indicating that LAGP has much better air stability than LLZO. Vacuum annealing at 350 °C hardly changes the composition of the surface. Therefore, we annealed the sample in 1×10^{-6} mbar O_2 at 350 °C for 30 minutes. As shown in the figure below, it can be seen that the surface C of the sample at 350 °C significantly decreases, while the signals of O and P are significantly enhanced. However, when the sample was cooled to room temperature, the signal of C increased significantly while the signal of O and P decreased. No such carbonization phenomenon was observed on the surface of LLZO.

Fig. S17. Evolution of surface species on LAGP during annealing and cooling processes.

We conducted APXPS on LAGP sample and the results are shown in the following figure. LAGP cannot react with CO₂ and H₂O even at a high pressure of 0.5 mbar. By comparing two electrolyte materials, we infer that the stability of LAGP comes from the hydrophobic carbon layer generated on the surface and the stability of PO₄ structure. Due to the stability of surface O, it is difficult for LAGP to directly react with CO₂ to generate Li₂CO₃. On the contrary, LLZO surface can react directly with CO₂ to form Li₂CO₃, which is a hydrophilic layer, may cause the occurrence of subsequent severe reactions. The results indicate that the hydrophilicity and hydrophobicity of the initial products on the clean electrolyte surface, as well as the stability of the surface O structure, may play an important role in determining the air stability of solid electrolyte materials.

We have added these comparisons in our manuscript in the “Comparison of LLZO with LAGP” part and added these two figures in Supplementary as Fig. 17 and Fig. 18.

Fig.18. *In situ* ambient pressure experiments on LAGP sample with CO₂ and H₂O.

Comments

1) The removal and re-deposition of surface Li_2CO_3 and LiOH at elevated temperatures and/or UHV have been reported by Refs. 15, 36, and R1.

Response: Thank you for your comment. The acquisition of clean LLZO surface is the basis for the following *in situ* experiment. We have gone through multiple explorations on the basis of the methods in these literatures to produce LLZO with clean surface. The O 1s XPS spectra in Ref.R1 indeed indicate that they produced a clean LLZO surface at 500 °C which is very similar with our results at 350 °C as shown in Fig.S3. The difference in temperature may come from the difference in sample or the position of thermocouples. The authors of Ref.R1 said that “the annealed LLZO data represent, to our knowledge, the only reported XPS analysis of LLZO without any interfacial surface reaction species.” Thus, this reference can also be used to support us that we obtained LLZO with a completely clean surface, we added the reference into our discussion to support our results as reference 27. In addition, the reaction mechanism of surface restoration by low-temperature annealing has not been discussed in the Ref.R1. We have provided a clear explanation of the mechanism using *in situ* XPS, XAS and mass spectrometry.

2) In Figure 4(a), a peak around 532 eV in O1s is attributed to carbonate. This peak is rather strongly observed in the spectra recorded under UHV (bottom). However, C1s XPS shows no (or very weak) peak in Figure 4(b). This discrepancy might be explained by assuming that the LiOH layer is formed after annealing as well as Li_2CO_3 . According to the databases of XPS, O1s peaks for metal hydroxides and metal carbonates appear around 531 eV. In addition, Li 1s for LiOH and Li_2CO_3 appears around 55 eV. Therefore, it is difficult to determine the surface species only from O1s and Li1s spectra. Furthermore, the MS data (Figure 2(e)) demonstrates that the base pressure of H_2O is higher than CO_2 , which also show the possibility of the re-deposition of LiOH . If the LiOH layer is thin (~ few nm), AP-mRAS might not clearly detect LiOH (Figure 3(c), (e)).

Response: Thank you for your comment. The most important reason for the mismatch between O and C peak intensities in Fig. 4 (now as Fig 5) is that the comparison signals come from different species, one is surface contaminated C, the other is lattice O. The amount of Li_2CO_3 on the surface after annealing and cooling is very small, the peak of surface contaminated C is so strong that it shows weak C 1s signals of carbonate. We used 650 eV for XPS detection, the detected O is very surface information, thus the O signal of Li_2CO_3 appears more obvious. The surface contaminated carbon can be removed by annealing in 1×10^{-6} mbar O_2 at 350 °C. As shown in the following flowchart, in order to reduce the impact of surface contaminated carbon on carbonate observation, we actually conducted 1×10^{-6} mbar O_2 annealing treatment for 30 min at 350 °C before the CO_2 experiment. After annealing in O_2 , the ratio of carbonate to surface

contaminated carbon changes significantly, while the peak of O 1s remains almost unchanged. The C 1s data we previously included in our manuscript in Fig 4 (now Fig.5) was the state after vacuum annealing (process (a) in the following figure), now we have changed it to the one after O₂ annealing (process (b) in the following figure). In addition, we have added a flowchart as Fig. 2 in the manuscript and a detailed description in the experimental section to make the phenomenon more clearly displayed.

The intensity difference between O 1s peak and C 1s peak also be affected by the atomic ratios and photoionization cross-sections: for Li₂CO₃ on the LLZO surface, the atomic ratios of oxygen and carbon are 3:1 and have different photoionization cross-sections (O is 0.3383 and C is 0.1308 at 650 eV). The calculated O/C area ratio of Li₂CO₃ is approximately 7.76:1 at 650 eV, so it appears that the peak of O is more pronounced than that of C.

The databases of XPS were measured under different conditions and can be affected by many factors, which can only serve as a reference. In order to increase the credibility of our results, we added *in situ* experiments to obtain accurate differences between pure LiOH and Li₂CO₃. As shown in the following figure, Li metal was *in situ* scraped using a Wobble-Stick with sharp blade surface. By utilizing near ambient pressure technology, we can obtain pure Li₂O, LiOH and Li₂CO₃. From O 1s spectra, it can be seen that the difference between Li₂CO₃ (~531.5 eV FWHM~1.75 eV) and LiOH (~530.9 eV FWHM~1.61 eV) is about 0.6 eV, which is very obvious and basically consistent with our peak fitting in Fig. 6.

By substituting the peak positions and half width obtained here into our CO₂ data fitting. We found that O 1s and C 1s peaks at different stages during annealing and CO₂ experiment can be well fitted with a Gaussian peak corresponding to CO₃²⁻. Then we calculated the peak area of O and C, the results are shown in the following table. We calculated the atomic ratio of O/C using the photoionization cross-sections of O (0.3383) and C (0.1308) at 650 eV. From the table, we can see that the O/C is maintained at a stable value ~3.9 during the pressure range 10⁻⁶-10⁻² mbar. **The stability of the ratio means that within this pressure range, the surface is relatively pure Li₂CO₃ without LiOH and the layer is very thin, and light can detect all information within the layer.** Compared with pure Li₂CO₃ ~2.48, the ratio is higher. This phenomenon can be explained simply by the following figure. When 650 eV X-ray is incident on the surface of the material, the photoelectron kinetic energy of O is about 120 eV, while that of C is 370 eV. The detected depth of C is larger, so for pure Li₂CO₃, the proportion of C will be higher.

For the sample after O₂ annealing, the ratio seems higher ~4.49 than Li₂CO₃ ~3.9, we think it may be caused by the following reasons: 1. The signal of CO₃²⁻ for annealed sample is too weak resulting in excessive fitting error. 2. There may be a very small amount of LiOH on the surface that comes from surface impurities, which can be ignored in peak fitting. **The O/C keeps stable at ~3.9 after introducing 1X10⁻⁶ mbar CO₂ means LiOH has completely disappeared, which will not affect our observation of the reaction between CO₂ and LLZO above 1X10⁻⁶ mbar.** For the results above 0.5 mbar CO₂, O/C is affected by the gas peak, the higher the CO₂ pressure, the lower the ratio. Interestingly, after we pumped 1 mbar CO₂ back to the vacuum, the O/C ratio slightly decrease to 3.75, indicating that the thickness of the Li₂CO₃ layer exceeded the detection depth of O photoelectrons. **We have added some explanation in the manuscript and added the following figure in the supporting information as Fig. 6.**

The base pressure of H₂O is higher than CO₂ is mainly caused by the different pumping ability and speed of molecular pump. It can be seen from Fig. 6 of the manuscript that the reaction pressure between H₂O and LLZO is much higher (over 1×10^{-4} mbar) than this value.

If LiOH cannot be observed through mRAS, then Li₂CO₃ produced by LiOH should not be observed either. However, after the reaction, we observed clear Li₂CO₃, indicating that the formation of Li₂CO₃ originated from the reaction between CO₂ and LLZO.

3) If LiOH is formed on the surface of annealed LLZO as indicated in the previous comment 2, the formation and/or growth of Li₂CO₃ layer (Figure 4(b)) is not the direct reaction of LLZO and CO₂, but the reaction of surface LiOH and CO₂.

Response: Thank you for your comment, in the answer to the comment 2, we have ruled out the influence of LiOH and confirm observed Li₂CO₃ comes from the reaction between CO₂ and LLZO.

4) In the Figure 4(d), a peak around 531 eV is assumed to be related to the Li extraction with references. Ref. 33 explains the O-K pre-edge peak is caused by the hybridization of transition metal d-orbital and oxygen 2p-orbital, and the peak shift is related to the change in the oxidation state of the transition metal. On the contrary, in this work, there is no element that changes its oxidation state, as it is assumed that lithium ions are extracted together with oxygen from the lattice.

Response: Thank you for your comment, O-K pre-edge peak is caused by the hybridization of transition metal d-orbital and oxygen 2p-orbital. In this work, there is no element that changes its oxidation state, which means the change of the valence state of oxygen with the removal of Li. When oxygen redox happen, changes in the pre-peak can be observed by XAS, which has been widely reported in cathode materials (Chem. Mater. 2019, 31, 19, 7864–7876, ACS Energy Lett. 2021, 6, 10, 3417–3424). For Li₂O and Li₂O₂, an increase in the valence state of O can cause the absorption spectrum peak position to shift towards the lower photon energy direction, as shown in the following figure *PLoS one* 7.11 (2012): e49182. If there is no significant change in the valence

state of O when Li^+/H^+ exchange occurs (shown in Fig. 6), there will be no changes in these peak shapes.

Minor comments

5) The specimens should be described in more detail. For example, where were the LLZO pellets polished? In a glovebox or in air?

Response: Thank you very much for your suggestion. We have greatly strengthened the detailed description of the experimental section, including sample preparation, polishing process and *in situ* measurements. All modifications are highlighted in the Methods section.

6) Composition of the solid electrolyte: which is true, Ta0.25 (in the abstract), or Ta0.5 (in the method section)?

Response: Thank you very much for your careful inspection. We have changed the components in the abstract to the correct “ $\text{Li}_{6.5}\text{La}_3\text{Zr}_{1.5}\text{Ta}_{0.5}\text{O}_{12}$ ”.

7) Standard binding energy of C1s XPS: which is true, 284.6 eV (line 121), or 284.8 eV (line 373)?

Response: Thank you very much for your careful inspection. We have changed the binding energy in the experimental section to the correct “284.6 eV”.

Reference

[R1] Zhu et al., *Adv. Energy Mater.* 2019, 9, 1803440.

REVIEWER COMMENTS

Reviewer #1 (Remarks to the Author):

The reviewer appreciated the revision & efforts that the authors made to strengthen the manuscript. The reviewer found the manuscript had much improved and support it be published.

However, considering the shared comments from all three reviewers regarding the novelty and new findings. The reviewer supposes it could be beneficial for the authors to provide a few paragraphs to emphasize/summarize the new finding that deepens previously understanding of LLZTO surface degradation. It will also be helpful if the authors also point out the new methodologies/characterization used in this research and emphasize the novelty in new instruments.

Reviewer #2 (Remarks to the Author):

Thank you for the corrections, improvements and additional experiments you made to adress my and the other reviewers comments! I think that this is now a strong publication, adding to the field and especially putting your method into the context of already existing methods.

I especially appreciate the additional experimental details given for sample preparation, as those are so crucial in LLZO research.

Reviewer #4 (Remarks to the Author):

The authors used DFT calculations to support their experimental results. They considered the LLZO (001) surface and calculated CO₂ and H₂O adsorption energies. Their results show that CO₂ has stronger adsorption energy (~ -6 eV) than H₂O (~ -1 eV) on the LLZO surface and based on this they conclude that CO₂ is more active than H₂O on the clean LLZO surface. They also calculated Li and O defect formation and the H⁺ and Li⁺ atoms exchange energies to confirm their experimental results. However, in my opinion, their theoretical calculations are trivial and appear to lack meaningful insights. For example, it's difficult to accept that the calculated CO₂ adsorption energy is in the range of -6 to -7 eV (Figure 7d), which is unusually high and potentially unrealistic. In general, the adsorption energy of gaseous species like CO₂ on surfaces is often in the few tenths to a few eV. I

believe this strange value is due to errors in their calculation or improper treatment of the system. They did not explicitly mention the inclusion of spin polarization and van der Waals interactions in their calculations, which is critical in this type of study.

Furthermore, there are concerns about their surface modeling. The chosen surface's stability is not clearly addressed. A comprehensive approach would include considering all possible terminations and selecting the most stable one. The lack of details on how this surface was determined raises concerns about the robustness and reliability of their modeling approach. Besides, it is important to confirm the adsorbed adsorbates on the adsorbents are minimum using vibrational frequency analysis, which is missing in this paper.

Lines 346-349 state that “our DFT results confirm that the reaction of CO₂ on the LLZO surface is thermodynamically preferred at low pressure due to the low absorption energy of CO₂, resulting in a concentration gradient of Li in the sub-layer.” The term “low absorption energy” is deceptive. According to their DFT results, CO₂ has a higher adsorption energy. Also, how do they relate adsorption energy with the concentration gradient of Li in the sub-layer?

Other minor comments are that authors should check the term ‘absorption energy’ or ‘adsorption energy’; the inappropriate use of the abbreviation LLZO in the abstract.

Given the aforementioned concerns, including the lack of meaningful insights, unusual calculated values, and methodological gaps, I question the conceptual merit of including this theoretical section in prestigious journals like Nature Communications. These concerns collectively cast doubt on the rigor and significance required for publication in such an esteemed journal.

Point-to-Point Response Letter to Reviewers' Comments

Dear Reviewers,

The authors greatly appreciate reviewers #1 and #2 for their affirmation of the revised manuscript, and to reviewer #4 for the valuable feedback on our DFT portion. Please find enclosed our point-to-point response to reviewers' comments and revised manuscript, which we would like to submit as the revised version of NCOMMS-23-29214A. **After in-depth discussion, we have decided to accept the suggestions of the editorial team to completely remove the DFT portion from the manuscript. The DFT portion is only a support for the experimental results, so deleting this part has almost no impact on the rest of the manuscript.** Changes made to the manuscript have been highlighted in PDF file. The point-to-point responses to reviewers' comments are as following.

Reviewer #1 (Remarks to the Author):

The reviewer appreciated the revision & efforts that the authors made to strengthen the manuscript. The reviewer found the manuscript had much improved and support it be published.

However, considering the shared comments from all three reviewers regarding the novelty and new findings. The reviewer supposes it could be beneficial for the authors to provide a few paragraphs to emphasize/summarize the new finding that deepens previously understanding of LLZTO surface degradation. It will also be helpful if the authors also point out the new methodologies/characterization used in this research and emphasize the novelty in new instruments.

Response: We thank the reviewer for acknowledging the acceptance of our work.

As suggested by reviewer, we have added a paragraph in the discussion section to highlight the new findings, new methodologies/characterization used in this research and emphasize the novelty in new instruments.

Reviewer #2 (Remarks to the Author):

Thank you for the corrections, improvements and additional experiments you made to address my and the other reviewers comments! I think that this is now a strong publication, adding to the field and especially putting your method into the context of already existing methods.

I especially appreciate the additional experimental details given for sample preparation, as those are so crucial in LLZO research.

Response: We thank the reviewer for acknowledging the contribution of this work and for the acceptance of our work. We truly appreciate the reviewer for his/her valuable comments and suggestions, which enormously improve the quality and clarity of this

manuscript.

Reviewer #4 (Remarks to the Author):

The authors used DFT calculations to support their experimental results. They considered the LLZO (001) surface and calculated CO₂ and H₂O adsorption energies. Their results show that CO₂ has stronger adsorption energy (~ -6 eV) than H₂O (~ -1 eV) on the LLZO surface and based on this they conclude that CO₂ is more active than H₂O on the clean LLZO surface. They also calculated Li and O defect formation and the H⁺ and Li⁺ atoms exchange energies to confirm their experimental results. However, in my opinion, their theoretical calculations are trivial and appear to lack meaningful insights. For example, it's difficult to accept that the calculated CO₂ adsorption energy is in the range of -6 to -7 eV (Figure 7d), which is unusually high and potentially unrealistic. In general, the adsorption energy of gaseous species like CO₂ on surfaces is often in the few tenths to a few eV. I believe this strange value is due to errors in their calculation or improper treatment of the system. They did not explicitly mention the inclusion of spin polarization and van der Waals interactions in their calculations, which is critical in this type of study.

Response: Thank you for your comments. Although we observed a very low reaction pressure of CO₂ on LLZO experimentally, we agree with the review's viewpoint that the calculated adsorption energy range of -6 to -7eV for CO₂ is indeed too high. We did not include the influence of van der Waals interactions in the calculation, which may have caused these deviations. **Therefore, in order to avoid the impact of calculation bias, we have decided to remove the entire DFT section from the manuscript including DFT portion and Fig.7 in main text and experimental section, and supplementary Fig.13-16. The DFT portion is only a support for the experimental results, so deleting this part has almost no impact on the rest of the manuscript.** The description of "adsorption energy" that appeared twice in other parts of the manuscript has also been deleted.

Furthermore, there are concerns about their surface modeling. The chosen surface's stability is not clearly addressed. A comprehensive approach would include considering all possible terminations and selecting the most stable one. The lack of details on how this surface was determined raises concerns about the robustness and reliability of their modeling approach. Besides, it is important to confirm the adsorbed adsorbates on the adsorbents are minimum using vibrational frequency analysis, which is missing in this paper.

Response: Thank you for your comments. Our termination selection is referred to previous articles on LLZO calculation, in which (001) orientations have been shown as the possible low-energy surface (ACS Appl. Mater. Interfaces 2021, 13, 44, 52629–52635, Chem. Mater. 2017, 29, 7961–7968, Nature Reviews Materials 5.2 (2020): 105-126, ACS Appl. Mater. Interfaces 2019, 11, 1, 898–905). Therefore, we did not consider

all possible terminations, which may indeed lead to inaccurate modeling approach.

Lines 346-349 state that “our DFT results confirm that the reaction of CO₂ on the LLZO surface is thermodynamically preferred at low pressure due to the low absorption energy of CO₂, resulting in a concentration gradient of Li in the sub-layer.” The term “low absorption energy” is deceptive. According to their DFT results, CO₂ has a higher adsorption energy. Also, how do they relate adsorption energy with the concentration gradient of Li in the sub-layer?

Response: Thank you for your comments. The misunderstanding of this sentence is due to our improper expression. What we mean is that CO₂ is more easily adsorbed on LLZO surface, which will attract internal lithium to the surface to participate in the reaction, leading to the appearance of lithium vacancies in sub-layer. **We have deleted all these sentences in the revised manuscript.**

Other minor comments are that authors should check the term ‘absorption energy’ or ‘adsorption energy’; the inappropriate use of the abbreviation LLZO in the abstract.

Response: Thank you for your comments. **We have deleted all “adsorption energy” in the revised manuscript.** It is very common to abbreviate Ta doped LLZTO as “LLZO” in LLZO related articles, and we consistently use the abbreviation LLZO in our manuscript.

Given the aforementioned concerns, including the lack of meaningful insights, unusual calculated values, and methodological gaps, I question the conceptual merit of including this theoretical section in prestigious journals like Nature Communications. These concerns collectively cast doubt on the rigor and significance required for publication in such an esteemed journal.

Response: Thank you for your comments. **We accepted your comments. We have removed the entire DFT section from the manuscript. Deleting this theoretical section has almost no impact on the rest of the manuscript.**